# Photochemical synthesis of natural lipids in artificial and living cells

Peng Ji ⓘ , Alexander Harjung ⓘ , Caroline H. Knittel, Alessandro Fracassi ⓘ , Jiyue Chen, Roberto J. Brea ⓘ & Neal K. Devaraj ⓘ ✉

Lipid synthesis plays a central role in cell structure, signaling, and metabolism. A general method for the abiogenesis of natural lipids could transform the development of lifelike artificial cells and unlock new ways to explore lipid functions in living cells. Here, we demonstrate the abiotic formation of natural lipids in water using visible-light-driven photoredox chemistry. Radical-mediated coupling of hydrocarbon tails to polar single-chain precursors yields lipids identical to those enzymatically formed. Spatiotemporally controlled lipid generation promotes de novo vesicle formation, growth, and division. Lipid synthesis can be driven by RNA aptamers that specifically bind and activate photocatalysts, establishing a direct link between abiotic lipid metabolism and nucleic acid sequence. Light-mediated assembly of bioactive lipids can take place in living cells, triggering signaling events such as apoptosis and protein kinase C (PKC) activation. Our finding that photochemical lipid synthesis can be driven by simple genetic elements could be the starting point for developing protocells capable of Darwinian evolution. Additionally, the ability to generate specific membrane lipids in living cells with precise spatiotemporal control will advance studies on how lipid structure influences cellular function.

Lipids are essential molecular components of all life on Earth. Membrane lipids define cellular boundaries, such as the plasma membrane, and were likely critical for early protocell compartmentalization[1,2]. Lipids play important roles in cell signaling, and their dysregulation often leads to disease[3]. Cells synthesize lipids enzymatically, with a key step frequently being the acylation of a polar head group with a fatty hydrocarbon tail, catalyzed by a membrane protein[4,5]. Despite advances over the last decades, reconstituting biochemical lipid synthesis ex cellulo remains challenging[6,7]. A general strategy for synthesizing the wide range of membrane lipids found in biology under physiologically relevant conditions would have several applications in generating lipid-based materials. For instance, developing lifelike artificial cells will likely require integrating lipid compartment metabolism with nucleic acid replication and natural membrane lipids are intrinsically compatible with the necessary biological machinery. Additionally, the ability to produce specific lipids in cells would enable direct elucidation of lipid structure-function relationships in biology.

As an alternative to enzymatic synthesis, several studies have attempted to generate natural lipids, or very close analogs, abiotically in water[8-14]. Previous approaches to abiotically generate natural lipids in water have used acylation chemistry to mimic the enzymatic synthesis of lipids[12-14]. For instance, recent work has shown that lysophospholipids can be acylated in alkaline media to create natural lipids[14], but this process requires several hours and provides high yield only under basic conditions. Under more physiologically relevant conditions such as in PBS (pH 7.4) or HEPES buffer (pH 7.5), the reaction yield was negligible (<1%), which makes this approach interesting for prebiotic chemistry, but limits its applicability to biological systems (Supplementary Fig. 1a). Another example of abiotic lipid synthesis involves a chemoselective strategy for in situ ceramide formation[13], which employs cell-permeable ligation partners to generate ceramides in PBS (pH 7.4, 37 °C). While this method operates under physiological conditions, ceramide formation requires several hours and the method is restricted to ceramide synthesis, limiting its broader utility

---

Department of Chemistry and Biochemistry, University of California, San Diego, La Jolla, CA, USA. ✉e-mail: ndevaraj@ucsd.edu

for the production of diverse lipid species (Supplementary Fig. 1b). In seeking a unified method to form a diverse array of natural lipids abiotically, we explored approaches distinct from the traditional acylation of polar head groups by activated fatty acids, as used both enzymatically and in previous synthetic methods. Since most lipids consist of hydrocarbon tails attached to a more polar functional group, a mechanism to directly ligate hydrocarbon tails to polar lipid precursors via carbon-carbon bond formation would be applicable for generating multiple natural lipids. Numerous studies have demonstrated that photoredox coupling chemistry is attractive for achieving carbon-carbon bond formation due to its mild nature[15], the potential for spatiotemporal control[16], and biocompatibility[17,18], even for applications with living cells[19–21]. Despite these significant advances, the generation of self-assembling phospholipid species in water through photoredox-mediated coupling remains unexplored.

Here we describe the photocatalytic synthesis of natural membrane lipids in water, leading to the spontaneous assembly of protocell vesicles. Photoredox lipid ligation (PLL) between *N*-hydroxyphthalimide (NHPI) fatty esters, one of the most extensively studied precursors for radical coupling[22,23], and olefin-modified lysolipids forms carbon-carbon bonds, generating natural lipids (Fig. 1a). Several lipid classes can be formed in water, including phospholipids (the main

structural components of cell membranes, with hydrocarbon tails appended through *O*-acyl alkyl tails), sphingolipids (highly enriched in the nervous system and involved in several biological processes and diseases[24], with an *N*-acyl alkyl tail), and diacylglycerols (key intermediates in lipid metabolism, bearing two fatty acyl chains on a glycerol backbone). The presence of unsaturation in the precursors does not interfere with the occurrence of PLL. Light catalyzed synthesis of phospholipids in water leads to the de novo formation of protocells with biologically identical lipids in their membranes. Continuous irradiation of lipid precursors causes vesicle growth, budding, and division (Fig. 1b). Various organic photosensitizers, including those activated by nucleic acids such as DNA and RNA aptamers (Fig. 1c), can trigger PLL. In the presence of living cells, we demonstrate assembly of bioactive signaling lipids like ceramides and diacylglycerols (Fig. 1d). PLL shows promise for creating more lifelike artificial cell membranes and as a tool for understanding the roles of specific lipids in living cells.

## Results and discussion
### Synthesis of natural lipids by photoredox lipid ligation (PLL)
Recent studies have demonstrated that the photoinduced oxidative decarboxylation of carboxylic acids can mediate radical carbon-carbon bond cross-coupling reactions in aqueous conditions. This

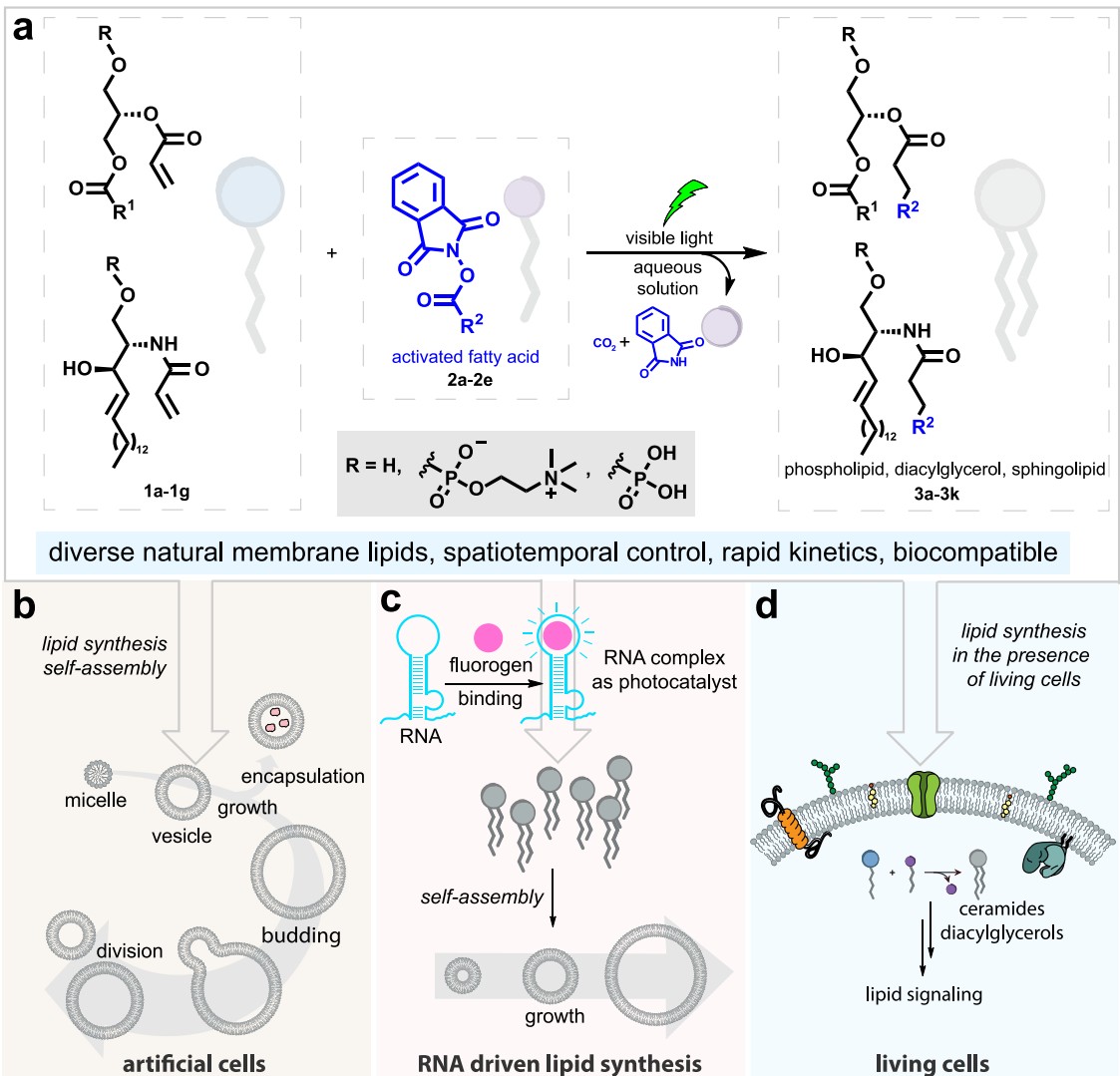

**Fig. 1 | Unified strategy for the synthesis of a wide range of natural membrane lipids in artificial and living cells through photoredox lipid ligation (PLL). a** Schematic illustration of PLL through photocatalytic decarboxylative conjugation. **b** PLL generates artificial cell membranes that grow and divide. **c** Fluorogen binding RNA aptamers can catalyze lipid synthesis through PLL. **d** PLL in living cells can generate specific bioactive lipids that trigger cell signaling.

method has been shown to have several promising applications in chemical biology, including DNA modification[25] and protein bioconjugation[17]. However, decarboxylative cross-coupling reactions often require oxygen-free conditions, are sluggish, and, for the generation of lipids, would require the use of fatty acids as precursors, which are not bioorthogonal. Indeed, we initially explored adapting such reactions for natural lipid synthesis and found that only trace quantities of the desired lipid products could be detected (Supplementary Table 1).

Rather than oxidative decarboxylation, we next turned to reductive decarboxylation. Since seminal work by Okada and coworkers[26], many groups have shown that photoredox driven radical cross-coupling can take place by single electron transfer (SET) to NHPI esters derived from carboxylic acids[22,23]. Reductive fragmentation leads to the extrusion of carbon dioxide and generation of a carbon centered radical that can be coupled to a Michael-type acceptor[23]. Recent studies[22,23] have demonstrated that this conjugation works in aqueous conditions and, depending on the choice of photocatalysts, can be conducted with visible light (>500 nm excitation)[27], which is an important prerequisite for biocompatibility. To adapt reductive decarboxylative coupling for PLL, we synthesized an NHPI ester (**2a**) from myristic acid and the acrylate derivative of oleoyl-lysophosphatidyl choline (**1a**). We formed a thin film of these precursors, hydrated the film with PBS buffer, and added 5 mol% of eosin Y as a photocatalyst (Fig. 2a). As a reducing agent, we used three equivalents of 1-benzyl-1,4-dihydronicotinamide (BNAH), a well-known 1,4-dihydronicotinamide adenine dinucleotide (NADH) analog. When this mixture was irradiated by 525 nm LED light for 30 minutes, the natural phospholipid 1-oleoyl-2-palmitoyl-*sn*-glycero-3-phosphocholine (OPPC, **3a**)[28] was readily detected by high performance liquid chromatography mass spectrometry (HPLC-MS), and an isolated product yield of 95% was obtained (Fig. 2b, entry 1).

We also explored alternative photocatalysts (Fig. 2b, Supplementary Table 2) and reducing agents (Fig. 2b, entry 1 and 4–8). Using rhodamine B as a photocatalyst did not significantly alter

yields (92%, excitation 525 nm, Fig. 2b, entry 9). We observed OPPC formation with reduced yield (77%) under the irradiation of blue light (450 nm) in the absence of eosin Y (Supplementary Table 3), consistent with previous work showing that BNAH can act as its own photocatalyst[15]. The choice of reducing agent was very important and switching BNAH with water soluble NADH (Fig. 2b, entry 6), ascorbate (Fig. 2b, entry 7), or *i*-Pr$_2$NEt (Fig. 2b, entry 8) only yielded trace lipid products, which might be attributed to the colocalization of BNAH with lipophilic precursors. Control experiments (Fig. 2b, entry 2–3; Supplementary Table 2, entry 6) confirmed that green light, photocatalyst, and reductant were all essential for PLL to take place. We also found that the reaction displays chemoselectivity despite the presence of NHPI esters, which have previously been used to activate carboxylic acids for coupling with amines[29]. Running the reaction in DMEM cell media with 10% fetal bovine serum, which contains a high concentration of amino acids (~6 mM), did not compromise reaction efficiency significantly (Fig. 2b, entry 10). In mixtures of **1a**, **2a**, and BNAH, we observed good stability of the NHPI ester over 10 minutes when incubated with 1 mM lysine, 0.5 mM sphingosine, or DMEM buffer (Supplementary Fig. 2). We hypothesize that the selectivity and high yield of reaction can be attributed to the self-assembling nature of the long-chain lipophilic reagents, which likely form structures such as micelles or emulsion droplets in which the reactive groups are positioned in close local proximity to one another and exclude highly polar reagents like amino acids. In contrast, if non-amphiphilic olefins are reacted, such as *N*-acryloylglycine, photo-driven ligation with NHPI ester **2a** did not take place (Supplementary Fig. 3). We also assessed the efficiency of the PLL reaction in the presence of lipid molecules commonly found in membranes, such as palmitic acid and cholesterol, and observed yields comparable to those obtained in the absence of additional lipids, further supporting the selectivity of PLL (Supplementary Table 4). The rapid kinetics of PLL (Supplementary Fig. 4) also likely contributes to the observed selectivity. Furthermore, performing the reaction in the presence of 40 mM SDS resulted in minimal product formation (less than 5%),

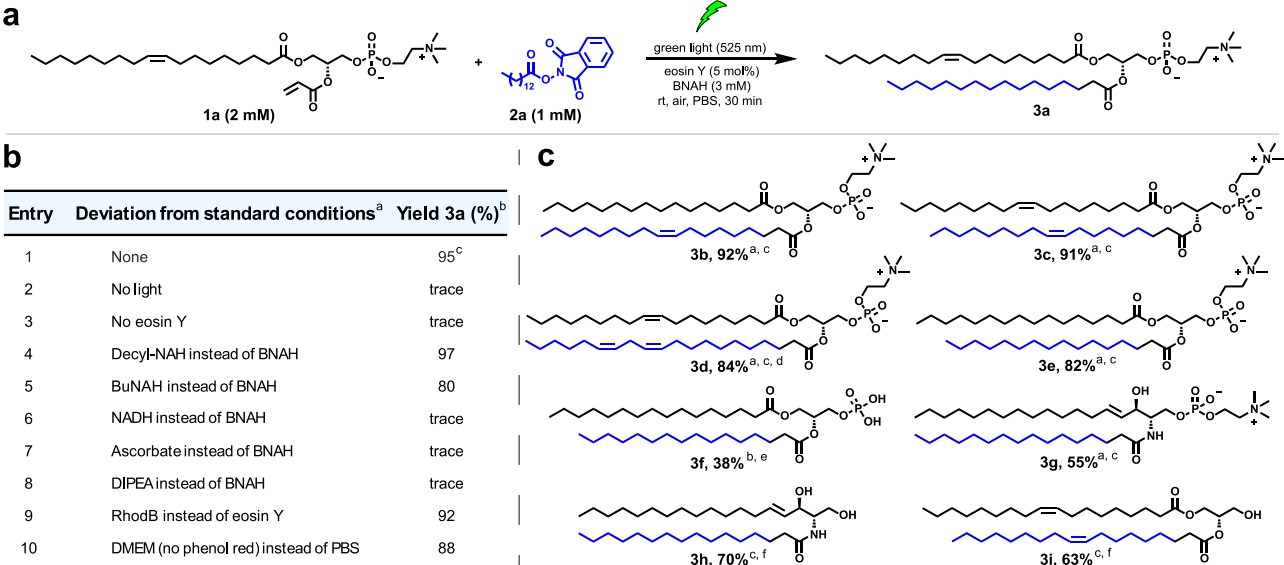

**Fig. 2 | Reaction condition optimization and selected scope of natural lipids synthesized by PLL. a** Reaction in PBS buffer between 18:1-lyso-PC-acrylate **1a** and NHPI ester **2a** under green light, generating natural phospholipid OPPC **3a**. The protocol shown represents the standard conditions. **b** Reaction condition optimization using different reductants and photocatalysts (and relative controls) for the synthesis of phospholipid **3a**. **c** Examples of natural lipids including phosphatidylcholines (**3b**, **3c**, **3d**, **3e**), phosphatidic acid (**3f**), sphingomyelin (**3g**), ceramide (**3h**), and diacylglycerol (**3i**) formed by PLL. Additional chemical structures can be

found in Supplementary Table 3. ᵃReaction conditions unless specified: a mixture of lysolipid **1** (2 mM), NHPI ester **2** (1 mM), BNAH (3 mM), and eosin Y (5 mol%) in PBS was irradiated by green light (525 nm) for 30 min at room temperature. ᵇHPLC yield. ᶜIsolated yield. ᵈReaction time is 15 min. ᵉThe reaction was performed without eosin Y under blue light (450 nm). ᶠ**1g** or **1f** (1 mM), NHPI ester **2** (2 mM), BNAH (3 mM), 1-oleoyl-2-hydroxy-*sn*-glycero-3-phosphocholine (18:1-lyso-PC) (3 mM), eosin Y (0.05 mM) in H$_2$O under green light irradiation (525 nm) for 30 min at room temperature.

Figure 2b table:

| Entry | Deviation from standard conditions[a] | Yield 3a (%)[b] |
|---|---|---|
| 1 | None | 95[c] |
| 2 | No light | trace |
| 3 | No eosin Y | trace |
| 4 | Decyl-NAH instead of BNAH | 97 |
| 5 | BuNAH instead of BNAH | 80 |
| 6 | NADH instead of BNAH | trace |
| 7 | Ascorbate instead of BNAH | trace |
| 8 | DIPEA instead of BNAH | trace |
| 9 | RhodB instead of eosin Y | 92 |
| 10 | DMEM (no phenol red) instead of PBS | 88 |

Structure labels in panel c: 3b, 92%[a, c]; 3c, 91%[a, c]; 3d, 84%[a, c, d]; 3e, 82%[a, c]; 3f, 38%[b, e]; 3g, 55%[a, c]; 3h, 70%[c, f]; 3i, 63%[c, f]

indicating that self-assembly is essential for efficient reaction progress (Supplementary Fig. 5).

There are multiple lipid classes in cells such as phospholipids, sphingolipids, and diacylglycerols[30]. The mechanism of PLL (Supplementary Fig. 6) suggests flexibility in the type of lipids that can be generated, provided that the target natural lipid possesses a fatty acyl tail. Changing the NHPI ester and the lipid olefin precursors enabled the synthesis of numerous phosphatidylcholines with either saturated (**3e** in Fig. 2c and **3k** in Supplementary Fig. 7), unsaturated (**3a** in Fig. 2a; **3b**, **3c** in Fig. 2c; **3j** in Supplementary Fig. 7) or polyunsaturated hydrocarbon tails (**3d** in Fig. 2c). The acrylate derivatives of lysophospholipids can be easily synthesized following established procedures (see Supplementary Information). While acylation typically proceeds efficiently, some lysophospholipids (**1b** and **1c**) may require prolonged reaction time. We were able to show that alternative lipid head groups could also be used. For instance, PLL yielded diacylglycerol (**3i** in Fig. 2c) and phosphatidic acid (**3f** in Fig. 2c) in aqueous conditions, albeit with lower yields compared to phosphatidylcholines. We attribute the lower yield of compound **3i** primarily to the limited water solubility of its precursors which, even with additional detergent, led to the formation of insoluble suspended particles with presumably less efficient reagent mixing. We believe the lower yield observed for phosphatidic acid **3f** may be due to the expected higher critical micelle concentration (cmc) of the 16:0 Lyso PA derivative (parent 16:0 Lyso PA cmc is 0.54 mM) compared to that of the 16:0 Lyso PC derivative (parent 16:0 Lyso PC cmc is 0.005 mM)[31], which may hinder co-localization of the starting materials in self-assembled structures. To generate sphingolipids, we synthesized the acrylamide-containing lipid precursors **1e** and **1f**. Using a similar PLL approach, we were able to synthesize sphingomyelin (**3g** in Fig. 2c), a major constituent of myelin sheaths, and ceramide (**3h** in Fig. 2c), a signaling sphingolipid involved in apoptosis[32]. The lower yields observed for sphingomyelin **3g** and ceramide **3h** may be explained by the lower electron deficiency of acrylamide-derived precursors compared to acrylate esters, which likely results in decreased reactivity, and limits their ability to capture the generated alkyl radicals.

## De novo vesicle assembly by PLL

Given that **3a** and **3b** (POPC) provided comparable synthetic yields, and that POPC is more prevalent in nature, we selected POPC for further characterization of PLL (Fig. 3a). HPLC-ELSD analysis (Fig. 3b) revealed that POPC formation in PBS under 30 minutes of green light irradiation (525 nm) yielded a clear product peak, whereas no product was detected in the dark. We then conducted kinetic studies under the same conditions, observing a rapid depletion of precursors **1b** and **2b** accompanied by a concomitant increase in POPC, reaching completion within 1 minute (Fig. 3c). These results collectively underscore the efficiency of the photochemical synthesis protocol employed for POPC. Based on previous studies[15,26], we estimate the quantum yield of PLL should be over 1, which possibly explains the high efficiency of the reaction. As lipid synthesis takes place in water (Fig. 3a–c), we suspected that PLL would lead to de novo protocell vesicle formation. Since lysophospholipids can act as vesicle disrupting agents, to generate vesicles using PLL we used a slight excess of the NHPI ester. A mixture consisting of **1b** (0.83 mM), **2b** (1.25 mM), BNAH (2.5 mM), and eosin Y (0.05 mM) was irradiated in PBS. As expected, neither **1b** nor **2b** on their own or combined formed giant vesicles (Supplementary Fig. 8), as determined by phase contrast light microscopy. Mixing all components of the reaction on a glass slide does not lead to observable vesicles if allowed to stand for 1 h in the absence of light (Fig. 3d middle). However, if the same mixture is exposed to green LED light for 1 h on a glass slide, large vesicles are readily observed under phase contrast microscopy (Fig. 3d, Supplementary Figs. 9 and 10), in agreement with our HPLC-MS data indicating that 1-palmitoyl-2-oleoyl-*sn*-glycero-3-phosphocholine (POPC) lipid synthesis takes place in high

yield. The shape of de novo formed vesicles on the glass slide are similar to vesicles formed by direct hydration of dried POPC (Supplementary Fig. 11). Interestingly, we also found that the fluorescent eosin Y is spontaneously incorporated on or within vesicle membranes during de novo vesicle formation (Fig. 3e, Supplementary Figs. 10 and 11). We next characterized the de novo formed lipid vesicles generated by PLL. CryoEM of POPC vesicles resulting from PLL revealed membrane bound spherical structures, with a membrane thickness (4.8 ± 0.1 nm) comparable to previous measurements of natural lipid membranes[33] (Fig. 3f, Supplementary Fig. 12). Additionally, PLL could take place in the presence of fluorescent proteins such as mCherry (Fig. 3g, h), and the proteins were spontaneously encapsulated within the vesicles formed.

We tested if de novo formed vesicles containing eosin Y remained catalytically active and could generate new lipids by PLL. After removing unbound eosin Y, vesicles containing spontaneously internalized eosin Y were exposed to all the PLL reactants except the eosin Y photocatalyst. Lysolipid and NHPI precursors (**1a** and **2b**) were selected to afford a lipid product different from the lipid used to form the initial vesicle population so it could be easily differentiated. After irradiation with green light (525 nm), 1,2-dioleoyl-*sn*-glycero-3-phosphocholine, **3c** (DOPC) synthesis was detected with 20% yield after 10 min (Supplementary Figs. 13 and 14). Similarly, the non-canonical phospholipid **3l** could also be synthesized in the presence of eosin Y containing vesicles (Supplementary Fig. 15). Thus, de novo formed protocells are themselves catalytic for additional lipid synthesis.

The light-driven nature of PLL offers the potential for spatio-temporal control over lipid synthesis. Microscope focused excitation of eosin Y (bandpass filter 530–580 nm) in a solution of NHPI ester **2b** and lysolipid **1b** on a glass slide to form POPC led to localized de novo vesicle formation (Fig. 3d, Supplementary Fig. 9, Supplementary Movie 1). Several morphological transformations were observed upon continued illumination, including vesicle tubulation, budding, and division (Fig. 3i; Supplementary Fig. 16; Supplementary Movies 1–3). These changes are likely due to additional lipid synthesis within the illumination region. Lipid precursors would be expected to continuously diffuse from the non-illuminated to the illuminated region. As new lipids are formed, expansion of preexisting membranes occurs. Controls without light or eosin Y photocatalyst did not yield de novo vesicle formation (Supplementary Movies 4 and 5) or vesicle growth (Supplementary Movie 6). Furthermore, we found that PLL allows de novo formation of membranes containing multiple lipid species from a mixture of saturated and unsaturated NHPI precursors, which closer mimic the complex mixture of lipids in living cell membranes. Mixing an equivalent amount of two different NHPI esters (**2a** and **2b**) with four equivalents of lysolipid (**1b**), followed by green light irradiation, resulted in the formation of vesicles composed of both 1,2-dipalmitoyl-*sn*-glycero-3-phosphocholine (DPPC) and POPC (1:1.7) (Supplementary Figs. 17 and 18).

## Nucleic acid driven PLL

It has been suggested that linking nucleic acid replication to lipid vesicle growth and division would offer a possible pathway for generating protocells capable of Darwinian evolution[1]. Thus, a long-standing goal in artificial cell research has been the discovery of mechanisms by which nucleic acids can be linked to protocell vesicle formation[34,35]. While previous studies have shown that the presence of DNA or RNA can affect lipid structures, we are not aware of mechanisms by which nucleic acids can catalyze the synthesis of lipids and subsequent protocell assembly. Since organic dyes such as eosin Y and rhodamine B can be used for PLL, we hypothesized that dyes that become activated in the presence of nucleic acids, such as intercalating dyes and aptamer binding fluorogens, might catalyze natural lipid synthesis by PLL in a manner that is dependent on the presence of specific nucleic acids. To test this hypothesis, we

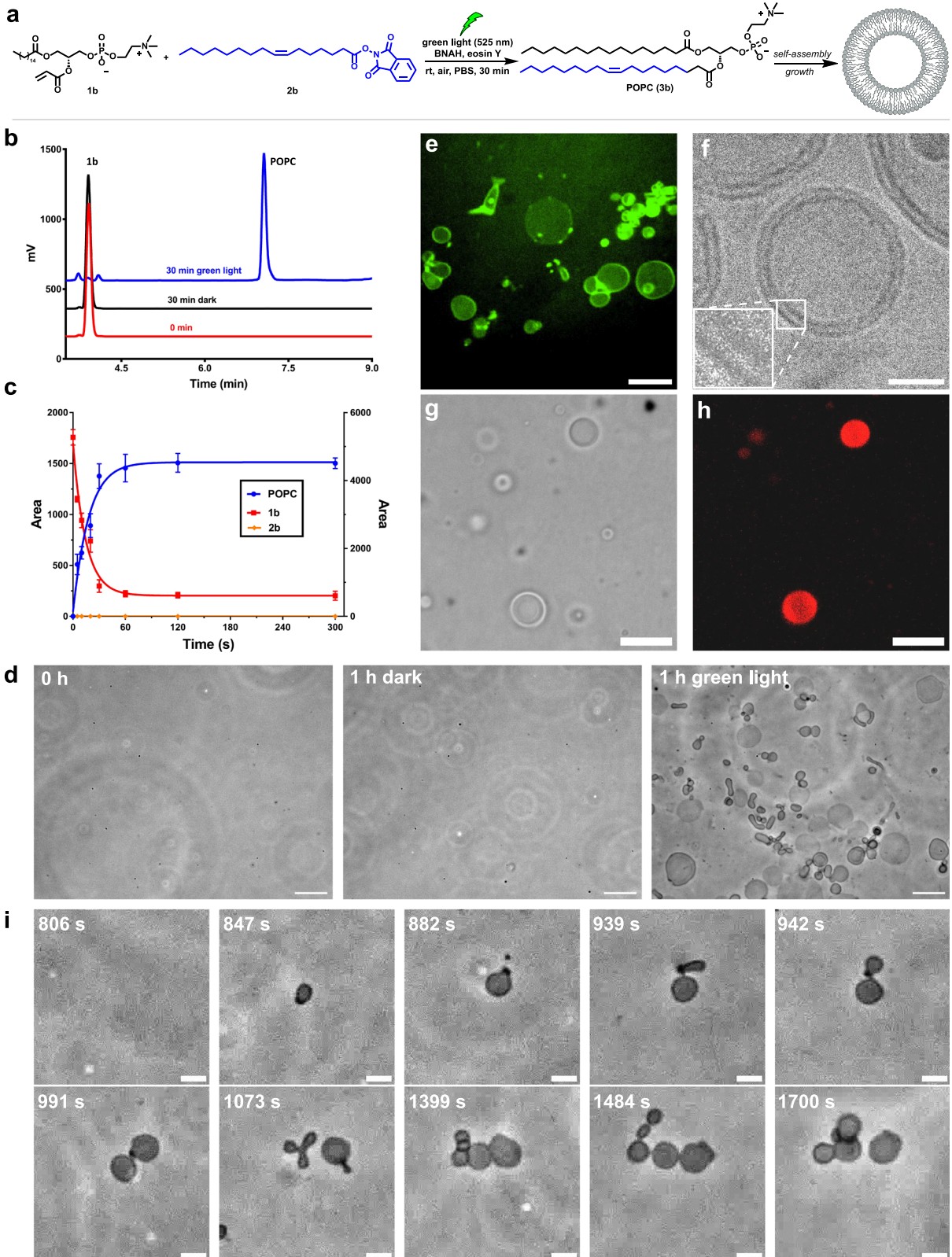

screened several DNA/RNA binding dyes for the ability to trigger PLL. A well-known class of activatable nucleic acid binding dyes are thiazole orange (TO) and its many related derivatives[36]. These dyes bind to nucleic acids, typically by intercalating between bases. Binding restricts conformational changes, preventing very rapid non-radiative decay and leading to activation[37]. We found that the DNA binding dye TOTO-1 (thiazole orange homodimer), when intercalated

into DNA, could act as a photocatalyst for the synthesis of POPC and trigger de novo vesicle formation (Supplementary Table 5, Supplementary Fig. 19). Furthermore, after diluting the newly formed vesicles 10-fold and imaging them by confocal microscopy, we observed fluorescence within the vesicles, demonstrating that a portion of the DNA is encapsulated during vesicle formation (Supplementary Fig. 19). This finding underscores how vesicles generated by PLL can

**Fig. 3 | De novo formation of protocell vesicles in water by PLL. a** Scheme depicting light-triggered reaction between lysolipid **1b** and NHPI ester **2b** forming POPC **3b** which self-assembles to form lipid vesicles. **b** HPLC-ELSD traces of POPC synthesis under green light. The reaction consisted of **1b** (2 mM), **2b** (1 mM), BNAH (3 mM), eosin Y (0.05 mM) in PBS for 30 min (blue line). In the absence of irradiation, no product was detected (black line). **c** Kinetics studies of POPC synthesis in PBS. Left y axis is the area of the ELSD signal of **2b** and POPC. Right y axis is the area of the ELSD signal of **1b**. Error bars represent standard deviation (SD). The data points are presented as means ± SD (*n* = 3 biologically independent samples). Source data are provided as a Source Data file. **d** Phase contrast microscopy images of localized de novo POPC vesicle formation on a glass slide before and after irradiation with green light for 1 h. Control in the absence of irradiation is also shown. The reaction mixture consisted of **1b** (0.83 mM), **2b** (1.25 mM), BNAH (2.5 mM), and eosin Y (0.05 mM) in PBS. Scale bar, 20 µm. **e** Confocal microscopy

image showing eosin Y localization during de novo POPC vesicle formation on the glass slide. Eosin Y appears to be localized on or within protocell vesicle membranes. Scale bar, 20 µm. **f** A representative cryoEM image of vesicles formed during the synthesis of POPC **3b** by PLL confirming the presence of a phospholipid bilayer structure. Scale bar, 20 nm. Bright field (**g**) and fluorescence (**h**) microscopy images demonstrate that in situ generation of POPC vesicles in the presence of fluorescent protein mCherry leads to spontaneous protein encapsulation. The unencapsulated protein was removed using Ni-NTA beads before imaging vesicles by confocal microscopy. Scale bar, 20 µm. **i** Time-lapse phase contrast microscopy images during de novo POPC protocell vesicle formation under localized green light excitation (530–580 nm). Vesicle growth, budding, and division events can be observed during irradiation. Scale bar, 5 µm. Representative images in (**d**–**h**) are shown from three independent experiments.

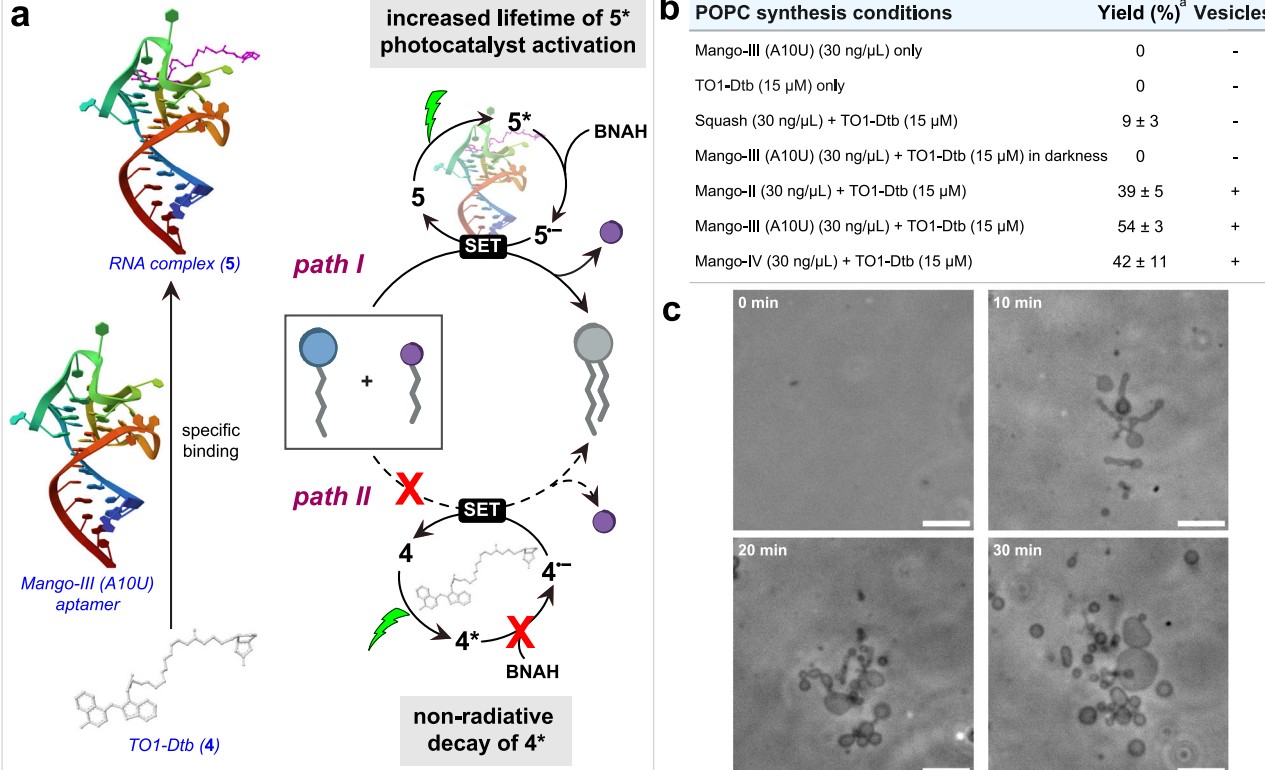

**b**

| POPC synthesis conditions | Yield (%)[a] | Vesicles |
|---|---|---|
| Mango-III (A10U) (30 ng/µL) only | 0 | - |
| TO1-Dtb (15 µM) only | 0 | - |
| Squash (30 ng/µL) + TO1-Dtb (15 µM) | 9 ± 3 | - |
| Mango-III (A10U) (30 ng/µL) + TO1-Dtb (15 µM) in darkness | 0 | - |
| Mango-II (30 ng/µL) + TO1-Dtb (15 µM) | 39 ± 5 | + |
| Mango-III (A10U) (30 ng/µL) + TO1-Dtb (15 µM) | 54 ± 3 | + |
| Mango-IV (30 ng/µL) + TO1-Dtb (15 µM) | 42 ± 11 | + |

**Fig. 4 | Nucleic acid directed natural phospholipid synthesis under green light irradiation. a** Mango-III (A10U) aptamer binds and activates TO1-Dtb, which can act as a photoredox catalyst for PLL. In the absence of the RNA aptamer, TO1-Dtb is deactivated and unable to effectively catalyze lipid synthesis. Schematic illustration of aptamer and binding ligand adapted from PDB ID: 6E8U[51]. **b** RNA aptamers that activate thiazole orange dyes catalyze the synthesis of POPC from **1b** (2 mM), **2b**

(1 mM), BNAH (3 mM), under green light irradiation in PBS buffer. The data points are presented as means ± SD (*n* = 3 biologically independent samples). Source data are provided as a Source Data file. **c** Time-lapse phase contrast microscopy images of localized de novo POPC vesicle formation and growth on a glass slide by PLL using Mango-III (A10U) (30 ng/µL)/TO1-Dtb (15 µM). Scale bar, 10 µm. [a]Yield is determined by HPLC-ELSD.

effectively compartmentalize essential biomolecules, a key requirement for life-like structural organization. In the absence of DNA, POPC formation was not observed (Supplementary Fig. 19). It has been suggested that early forms of life likely used RNA to both store information and catalyze chemical reactions. TO derivatives have also been shown to be activated by RNA aptamers. Unlike DNA intercalators, binding is highly dependent on the specific sequence and structure of the RNA[38]. For instance, TO1-3PEG Desthiobiotin (TO1-Dtb) binds to and is activated by the evolved RNA Mango aptamers in a sequence dependent manner[39,40]. Similar to our results with DNA, we found that lipid synthesis and de novo vesicle formation could be catalyzed by PLL in the presence of various versions of the Mango RNA aptamer and TO1-Dtb dye. Yields were dependent on the specific aptamer sequence and correlated with the ability of the aptamer to enhance TO1-Dtb fluorescence, with the highest yield

obtained using the Mango-III (A10U) aptamer and TO1-Dtb dye (Fig. 4a–c, Supplementary Fig. 20). Vesicle formation was triggered by localized visible light excitation, leading to growth and budding events similar to our previous observations using other photoredox catalysts (Fig. 4c and Supplementary Fig. 21). No lipid synthesis occurred in the presence of either aptamer or dye alone (Fig. 4b, Supplementary Fig. 20). To further explore sequence specificity, we tested the Squash aptamer, which has been evolved to bind an alternative dye. The Squash aptamer was found to slightly enhance TO1-Dtb dye fluorescence, and we observed minimal lipid synthesis and no vesicle formation (Fig. 4b). These findings suggest the PLL provides a mechanism for linking lipid metabolism and nucleic acid replication in a sequence dependent manner. We are currently determining whether PLL can be driven by alternative dye activating nucleic acids.

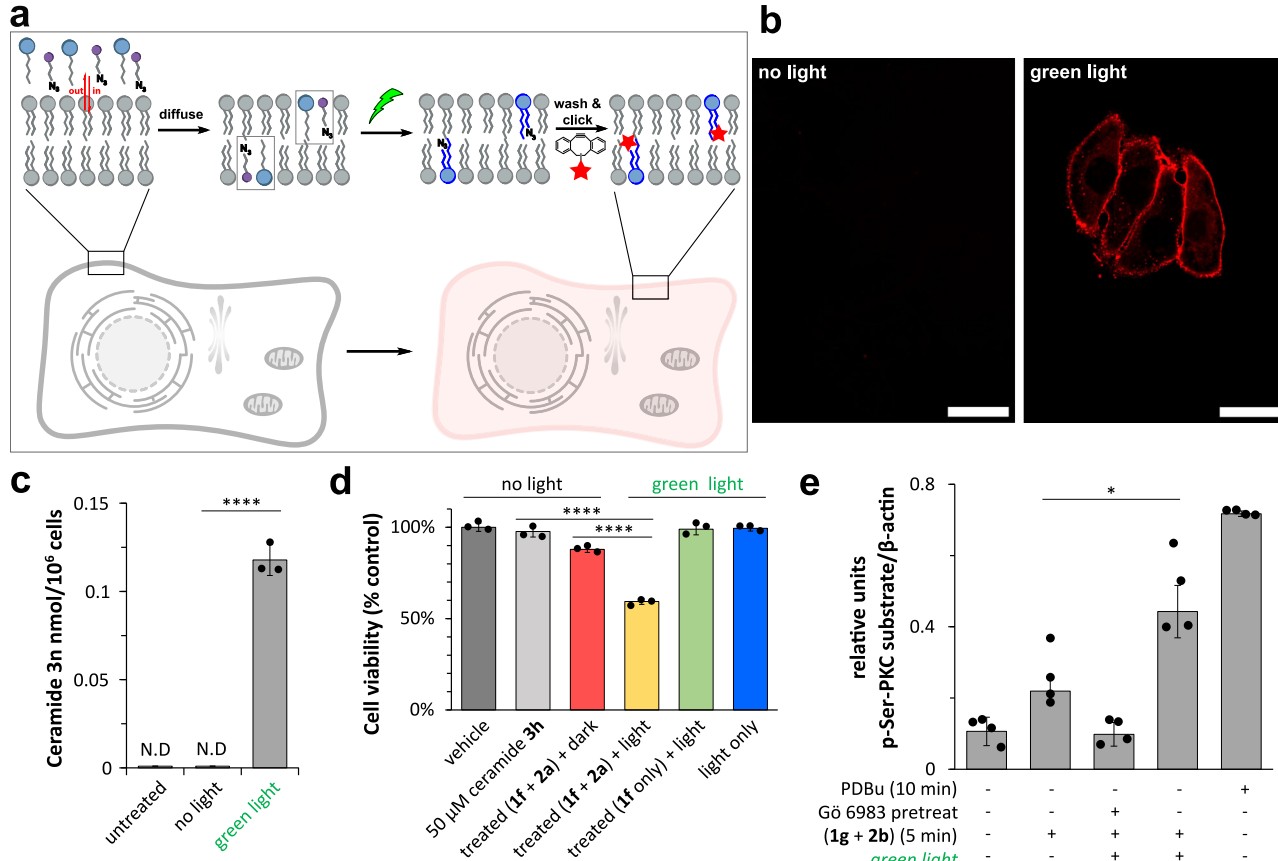

**Fig. 5 | Application of PLL in the presence of living cells. a** Schematic illustration of PLL in cells. **b** Confocal images of HeLa cells after treatment with **1b**, azido-NHPI ester **2e**, BNAH, and eosin Y under darkness (*left*) or by irradiation with 525 nm green light (*right*) for 5 min. After washing, HeLa cells were treated with DBCO-Fluor 594, washed, and then imaged by confocal microscopy. Scale bar, 50 μm. **c** Quantitative HRMS analysis of ceramide-$d_{27}$ synthesis in cells by PLL. HeLa cells were treated with 100 μM **1f**, 200 μM **2d**, 300 μM BNAH, and 5 μM eosin Y in DMEM followed by irradiation with green light for 5 min. The data points are presented as means ± SD ($n$ = 3 biologically independent samples). **d** Cell proliferation assay (CCK-8) of HeLa cells after light-controlled synthesis of ceramide. HeLa cells were treated with 50 μM **1f**, 100 μM **2a**, 150 μM BNAH, and 2.5 μM eosin Y in DMEM followed by irradiation with green light for 5 min. The data points are presented as means ± SD ($n$ = 3 biologically independent samples). **e** Western blot assay analyzing the relative amount of phosphorylated-Ser PKC substrates between 63 and 75 kDa. PLL synthesis of 1-2-dioleoyl-*sn*-glycerol leads to the selective activation of the PKC pathway in HeLa cells. HeLa cells treated with 25 μM **1g**, 50 μM **2b**, 75 μM BNAH, and 1.25 μM eosin Y in DMEM. The data points are presented as means ± SD ($n$ = 4 biologically independent samples). Statistically significant differences in (**c**–**e**) are indicated based on an independent t-test (two-tailed). $p$ = 0.00002 in (**c**); $p$ = 0.00004 in (**d**) for '50 μM ceramide **3h**' vs 'treated (**1f** + **2a**) + light'; $p$ = 0.00003 in (**d**) for 'treated (**1f** + **2a**) + dark' vs 'treated (**1f** + **2a**) + light'; $p$ = 0.015 in (**e**). $*p < 0.05$, $**p < 0.01$, $***p < 0.001$, $****p < 0.0001$. N.D. not detectable. Source data are provided as a Source Data file.

## PLL in living cells

Mammalian cells enzymatically generate thousands of unique lipids[41]. A major challenge has been deciphering not just the identity of specific lipids but determining how specific changes in chemical structure affect their biological function. Dysregulation of lipid metabolism often leads to the accumulation of specific toxic lipids and disease, but the molecular mechanisms of lipid toxicity are often obscure[42]. The ability to generate specific lipids in living cells would provide a tool for improving our understanding of lipid structure-function relationships. Based on the observed selectivity of PLL, and previous studies showing cell compatibility for photocatalysis using eosin Y[43], we speculated that PLL might be capable of synthesizing specific lipids in the presence of living cells. To initially determine live cell compatibility, we designed a simple experiment by which cells (HeLa) were exposed to synthetic NHPI lipid ester **2e** bearing a terminal azide group along with acrylate modified lysophosphatidylcholine **1b**. Cells were also exposed to eosin Y photocatalyst and the reducing agent. If synthesis of azido-phospholipid **3m** (Supplementary Fig. 22) takes place by PLL, the product phospholipid would be expected to be more easily retained in cell membranes compared to the single-chain precursors. After a

washing step, the presence of azido-phospholipids in the membrane was monitored by labeling of the azide groups with a fluorescent cyclooctyne dye (Fig. 5a). Cells were treated with lipid precursors, irradiated with 525 nm LED light for 5 min, and washed with cell media. Subsequent staining with cyclooctyne DBCO-Fluor 594 revealed significant membrane modification (Fig. 5b). To ensure the robustness of our findings, we performed several control experiments, including imaging untreated cells (Supplementary Fig. 22a), treated cells in the absence of light (Fig. 5b (*left*) and Supplementary Fig. 22b), and treated cells with all the necessary components for PLL under green light (525 nm), except for lysolipid **1b** (Supplementary Fig. 22c), or photocatalyst (Supplementary Fig. 22d). In all controls, significantly less staining was observed (Fig. 5b (left) and Supplementary Fig. 22), likely due to the ability of the single-chain precursors to be washed out more readily than the two-chain phospholipid product. Furthermore, we conducted long-term viability studies by varying eosin Y concentrations and green light intensity, observing in both cases negligible cytotoxicity under the PLL conditions (Supplementary Fig. 23).

Having tested cell compatibility, we next sought to determine if PLL could generate bioactive lipids in living cell membranes. We initially

tested the synthesis of ceramide, which is a central molecule in sphingolipid biosynthesis and has been shown to play a key role in apoptosis[44,45]. While ceramides and synthetic analogs have been extensively studied, the poor solubility of native N-acyl ceramides renders them unable to cross the cell membrane and has impeded studies on how the chemical structure of ceramides affects their cellular function. We designed a deuterated NHPI ester **2d** that can be coupled to acrylamide modified sphingosine **1f** via PLL to generate an isotopically labeled ceramide species **3n**. Deuterated ceramide can be differentiated from native ceramide by quantitative high-resolution mass spectrometry. We performed PLL to synthesize deuterated ceramides in HeLa cells, followed by a total lipid extraction and analysis of product formation by mass spectrometry, which revealed the formation of physiological quantities of ceramide[13] (Fig. 5c and Supplementary Fig. 24). Furthermore, photogeneration of ceramide led to a reduction in cell viability compared to several controls and in agreement with previously reported results[13] (Fig. 5d). Past work has shown that sphingosine itself can trigger apoptosis. However, we found that the acrylamide-modified sphingosine **1f** showed reduced toxicity (Supplementary Fig. 25), highlighting an advantage of using abiotic precursors. The external addition of ceramide did not reduce cell viability (Fig. 5d), as expected given its poor membrane permeability. The lack of activity from externally added ceramide also suggests that the photochemical synthesis of ceramide species takes place within cells, aided by the improved cellular permeability of the single-chain precursors.

As previously discussed, PLL can also be used to synthesize natural diacylglycerols (DAGs), which comprise an important class of lipid signaling molecules. DAGs activate enzymes in the protein kinase C (PKC) family, leading to phosphorylation of a wide range of downstream protein targets. Recent work has suggested that the location, timing, and structure of diacylglycerols significantly affect downstream signaling events and their impact on cell function[46]. We performed PLL to form 1-2-dioleoyl-sn-glycerol **3i** in the presence of HeLa cells and monitored the phosphorylation of protein targets of PKC adapting a previously developed western-blot assay[47]. Light-mediated formation of diacylglycerol **3i** led to significant upregulation of target protein phosphorylation (Fig. 5e and Supplementary Fig. 26), approximately 62% of that obtained using PKC activator phorbol 12,13-dibutyrate (PDBu) as a positive control (Supplementary Fig. 26). PKC inhibition using 1 μM Gö 6983, a pan PKC inhibitor, dramatically decreased the observed phosphorylation[48] (Fig. 5e). Running the reaction protocol while avoiding light significantly decreased phosphorylation (Fig. 5e). However, we did observe PKC activation above background, likely due to the acrylate monoacylglycerol precursor **1g** (Supplementary Fig. 26) and any inadvertent exposure to light during cell handling. Overall, our results suggest that PLL can be used to generate a variety of lipids in living systems and creates several opportunities for studying specific lipid species and their effect on cellular function.

We have demonstrated the aqueous synthesis of natural membrane lipids using photoredox mediated radical-driven carbon-carbon bond formation between hydrocarbon tails and polar head groups. Using visible light, we demonstrate de novo generation of 11 different lipids across 3 distinct membrane lipid classes. By using organic photocatalysts that are activated by nucleic acids such as RNA aptamers, we reveal a unique mechanism linking RNA sequence and structure to abiotic lipid metabolism. Furthermore, lipid synthesis is compatible with living mammalian cells and can be used to generate bioactive lipids and observe their effects on cell signaling. Our approach enables spatiotemporal control over lipid synthesis, paving the way for integrating RNA/DNA replication with artificial cell growth to create self-reproducing compartments capable of Darwinian evolution.

While this study demonstrates the efficiency of PLL as a strategy for de novo membrane formation, one key aspect that remains to be explored is whether the encapsulation of the catalyst within vesicles can effectively trigger the reaction from inside the compartments. This would be a crucial step towards developing autonomous synthetic cells capable of internally regulating membrane synthesis and remodeling. Further studies are needed to assess the feasibility of this approach and to investigate the functional properties that may emerge from such an ability. Additionally, future efforts should focus on identifying alternative photocatalysts or reaction pathways that can be activated at higher wavelengths, such as in the near-infrared range, to further enhance the biocompatibility of PLL and possibly open up in vivo applications. Due to synthetic constraints, we only focused on a subset of common phospholipids (e.g. phosphocholine, phosphatidic acid). Expanding this approach to include phospholipids with inositol, serine, or ethanolamine headgroups is an important direction for subsequent work, and we aim to investigate strategies to overcome the associated synthetic challenges. Other than nucleic acid binding dyes, we anticipate the employment of photoactivable metal-organic frameworks or nanocatalysts with tunable structures and large surface areas can potentially improve the reaction efficiency for some challenging substrates (**3f**) or more complex lipid synthesis[49,50]. However, considering the unknown metabolic stability of PLL precursors, it may be challenging to apply PLL for animal model studies, though aforementioned development of near-infrared photocatalysts and more stable PLL precursors would facilitate studies in more complex biological systems.

## Methods

### General procedure for the photochemical synthesis of natural phospholipids

To a 1-dram glass vial, 100 μL of a 10 mM stock solution of compound **1** (2.0 equiv) in CHCl₃, 50 μL of a 10 mM solution of compound **2** (1 equiv) in CHCl₃, 150 μL of a 10 mM BNAH (3.0 equiv) in CHCl₃, and 2.5 μL of a 10 mM solution of eosin Y (0.05 equiv) in MeOH were added. The solvent was evaporated under a gentle stream of N₂, and the content was redissolved with 300 μL of CH₂Cl₂. The CH₂Cl₂ was evaporated again, while carefully rotating the vial to obtain a thin lipid film. The film was hydrated with 500 μL of phosphate buffered saline and the suspension was sonicated for 2 min. The sample was irradiated with 18 W green LEDs ($\lambda_{max}$ 525 nm) positioned at approximately 1 cm distance for 30 min. The crude mixture was purified by silica gel column chromatography to provide the phospholipid product. Full experimental details and characterization of the newly described compounds are provided in the Supplementary Information.

### In situ POPC vesicle formation on a glass slide

The in situ synthesis of POPC was performed using reactants as described above. The microscopy samples were prepared by depositing 1–2 μL of the reaction mixture on a microscope glass slide after which a coverslip was added. The reaction was then imaged by phase contrast microscopy. The light required for PLL was provided by the microscope excitation source passed through a bandpass filter (excitation wavelength 530–580 nm). Images were collected at 0 min, 15 min, 30 min, and 60 min at the same position on the slide.

### In situ POPC vesicle formation using a fluorogenic nucleic acid binding dye

A lipid film containing 0.2 μmol **1b**, 0.1 μmol **2b**, 0.3 μmol BNAH was prepared and hydrated with a 100 μL PBS buffer (pH = 7.4) containing either 15 μM TOTO-1 and 30 ng/μL DNA or 15 μM TO1-3PEG-Desthiobiotin and 30 ng/μL RNA aptamer. The suspension was sonicated for 1 min, until a homogenous lipid dispersion was formed. The vial was irradiated with green LEDs ($\lambda_{max}$ = 525 nm) at approximately 1 cm distance for 30 min, while being cooled by an electronic fan. After 30 min, 20 μL of the solution was mixed with 80 μL MeOH and analyzed by HPLC-ELSD-MS.

### HeLa cell membrane labeling by PLL

HeLa cells (ATCC, CCL-2.2), maintained in DMEM (10% FBS, 1% penicillin/streptomycin), were plated in a 33 mm dish. After 24 h of

incubation at 37 °C, 5% $CO_2$, the cell medium was discarded. The cells were treated under the conditions specified in the Supplementary Information. Subsequently, the medium of the dishes was discarded, and the cells were washed with phenol red free DMEM three times. The HeLa cells were incubated with 0.5 mL of a 20 µg/mL solution of DBCO-Fluor-594 in DMEM for 10 min. Afterwards, the cells were washed with DMEM three times and then imaged by confocal microscopy. Full experimental details are provided in the Supplementary Information.

## HRMS quantification of ceramide synthesis in live cells

HeLa cells, maintained in DMEM were plated in triplicate using 6 cm culture dishes and grown to confluency at 37 °C, 5% $CO_2$. Once confluent, the medium was discarded and then the appropriate volume of DMEM was added. The cells were treated under the conditions specified in the Supplementary Information. After incubation, the media was removed, and the cells were washed 2 times using HBSS. Finally, 1 mL HBSS was added, and the cells were detached from their culture dish using a plastic cell scraper. The cells were counted before the lipid extraction (usually $2 \times 10^6$ cells/mL). Lipids were extracted from the cell suspension using the Bligh and Dyer method. The resulting residue was dissolved in 1 mL MeOH/CHCl₃ (9:1) and analyzed by HPLC-HRMS in the dark. Quantification of ceramide-d₂₇ (**3n**) was performed in triplicate using the software Tracefinder™ (ThermoScientific) and previously synthesized **3n** as an external standard. Full experimental details are provided in the Supplementary Information.

## Cell proliferation assay

HeLa cells were plated in a 96-well plate at a density of 10,000–20,000 cells/well. After 24 h of incubation at 37 °C, 5% $CO_2$, the medium was discarded. The cells were treated under the conditions specified in the Supplementary Information. All the six groups were incubated at 37 °C, 5% $CO_2$ for 24 h after treatment. After 24 h, 10 µL of Cell Counting Kit-8 solution was added to each well of the plate. The plate was incubated for 1 h in the cell incubator. The absorbance was measured at 450 nm using a Safire II plate reader (Tecan). All conditions were tested in triplicate and plotted with error bars representing standard deviation. Full experimental details are provided in the Supplementary Information.

## Western blot assay

HeLa cells were treated under the conditions specified in Supplementary Information. After the indicated treatment times, the cells were washed 2 times with HBSS before cell lysis. Cell lysates were prepared in lysis buffer directly after cell treatment. Cells were detached with a cell scraper and sonicated with a tip sonicator. The resulting cell lysates were separated on a Mini-PROTEAN® TGX™ gel (4-20%) and subsequently transferred to a PVDF membrane using the Trans-Blot® Turbo™ system. Membranes were blocked for 1 h at room temperature (5% BSA in TBS-T) after which they were incubated with primary antibody: Anti-Actin antibody (1:1000) (#3700, Cell Signaling Technology), Anti-Phospho-(Ser) PKC substrate antibody (1:1000) (#2261, Cell Signaling Technology). Following incubation overnight at 4 °C, membranes were washed with TBS-T and the corresponding secondary antibody was added for 1 h at rt. After five successive washes with TBS-T, blots were imaged with Thermo Scientific™ SuperSignal™ West Pico PLUS on a ChemiDoc™ XRS+. Full experimental details are provided in the Supplementary Information.

## Reporting summary

Further information on research design is available in the Nature Portfolio Reporting Summary linked to this article.

## Data availability

All data are available in the main text, supplementary materials and from the corresponding author(s) upon request. Source data are provided with this paper.

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

## Acknowledgements

The authors acknowledge the facilities along with the scientific and technical assistance of the staff of the cryoEM facility at UCSD. The UCSD molecular mass spectrometry facility is acknowledged. We also acknowledge helpful discussions with Prof. Itay Budin and Prof. Nathan Romero. This work was funded by the Department of Defense (N00014-22-1-2800 ONR VBFF), the National Institutes of Health (R35GM141939) and the Alfred P. Sloan Foundation (2022-19397).

## Author contributions

Conceptualization: P.J. and N.K.D. Methodology: P.J. Investigation: P.J., A.H., C.K., J.C., and R.J.B. Visualization: P.J. Funding acquisition: N.K.D. Supervision: N.K.D. Writing – original draft: P.J. and N.K.D. Writing – review & editing: P.J., A.F., A.H., C.K., J.C., N.K.D.

## Competing interests

The authors declare no competing interests.
