## [Peer Review file · Nature Communications]

Photochemical synthesis of natural lipids in artificial and living cells

Corresponding Author: Professor Neal Devaraj

Version 0:

Reviewer comments:

Reviewer #1

(Remarks to the Author)

Ji et al. report the development of a photoredox system for assembly of lipids under biocompatible conditions. Their method is called Photoredox Lipid Ligation (PLL) and enables the spatiotemporal control of lipid synthesis. This is achieved via visible light-mediated photoredox activation of N-acyloxy phthalimide esters using eosin Y as a photocatalyst for radical mediated coupling of hydrocarbon tails to polar single chain precursors. The resulting products are newly generated biologically relevant lipids. The authors first demonstrate the broad applicability of this method by synthesizing several different lipid products that vary in lipid tail and head group showcasing the versatility of this method. The authors next show the ability to generate vesicles from these photocatalytically-generated lipids and then follow this up with the use of DNA/RNA-nucleic acid binding dye conjugates that can achieve similar lipid products as their eosin Y method. Finally, the authors apply this labeling approach in living cells by assembling lipid products such as azide-labeled phospholipids, ceramide, and diacylglycerols and monitored the implications of these newly synthesized lipids visually and functionally through cell killing and/or protein activation. In the process, the authors also show that the lipid pre-cursors they use limit unwanted side effects compared to native precursors (e.g. acrylamide-modified sphingosine vs sphingosine) and/or the products they generate can achieve cell permeability in cases where the native products themselves cannot (in the case of ceramide). This highlights the potential utility of PLL for being able to better understand what lipids are doing in living cells without hindrance of background activity or lack of access to native environments.

Overall, this is an exciting work in the field of lipid biology. The ability to generate lipid products in live cells with exquisite spatiotemporal control and monitor their biological functions is an important methodology needed for better understanding how lipids work. Investigation into protein and sugar biomolecule function have long benefited from such tools and our ability to understand their biological functions in isolation or in the larger context of the surrounding cellular environment have greatly benefited as a result. This PLL approach further opens the door for similar studies to be performed on lipid biomolecules. In addition to basic biology pursuits, the ability to generate vesicles with this approach and study impact of lipid content on vesicle shape, structure, dimension could be a huge boost to building artificial vesicles with physiological/therapeutic importance.

Below are some comments to address for helping to improve the manuscript.

1. It could be beneficial to show an image or schematic in the main or supporting text that highlights the coupling mechanism. While N-acyloxy phthalimides have been widely used in other synthetic contexts, it would be helpful to introduce how that approach is being leveraged here. In particular, it would be helpful to the reader to highlight how this method compares to current/conventional in vitro lipid synthesis approaches (e.g. a figure showing past or previous assembly methods vs. the assembly method in this new work).

2. Lines 90-93. What is referred to by recent studies? Is this referring to the paper cited in previous sentences or other papers. Would be good to clarify.

3. Figure 2 panel b. What reaction yields does the condition optimization screen connect to? The yields for 3a? Would be good to more clearly highlight this in the panel and legend.

4. Have the authors tested the reaction in the presence of free fatty acids or other lipid-like molecules to get a sense of the reaction selectivity in the presence of other lipid biomolecule/building blocks? Presumably the reaction would be very selective, but might be good to showcase this with an in vitro experiment as part of the build up story to the utility of this method.
5. To address the high-yield being driven by product self-assembly, can the authors also perform the reaction in the presence of a suitable detergent to show that disruption or prevention of micelles/emulsion droplets prevents high yields?
6. Figure 2c. Can the authors address why the lower yields for Compounds 3f, 3g, and others in the text.
7. Line 174. Figures 3a-3c are very quickly glossed over in the text. Would be helpful to the reader to discuss these panels and the kinetics of reaction.
8. Regarding kinetics, can the authors gage how quickly the vesicles form after synthesis? Is this on par with typical/physiological vesicle formation kinetics? Along these lines, how does vesicle formation and shapes in Figure 3d and 3i compared to formation of vesicles from a native lipid precursor that is simply added together to form vesicles. This would be useful to get a sense of how different or similar vesicle formation is to normal vesicle formation through this synthetic approach. Furthermore, are the interesting vesicle shapes and structures generated through this approach unique to this method? If so, this could possibly open new avenues for spatiotemporal control of vesicle shapes and/or other biological/therapeutic applications.
9. Figure 3. The back and forth between POPC and OPPC is a bit difficult to follow in the text. Suggest to either stick with one vesicle type in the figure or more clearly call out the vesicle types in the figure images and/or provide more clear rationale or commentary for why moving between POPC and OPPC.
10. This reviewer found it difficult to follow the progression from figure 3 to figure 4 and then to figure 5. Seems like there is a more natural progression from figure 3 to figure 5 and that, by inserting figure 4 in between these two sections, there seems to be a deviation from the overall story. The use of DNA/RNA-photocatalyst complexes is an interesting concept to induce lipid and vesicle formation but comes off as just another type of photocatalyst that can be used to generate lipids through this approach rather than an advancement on origin of life study capabilities. As a suggestion to help with the flow of this section, perhaps the authors can determine whether DNA can be encapsulated within the newly generated vesicles similar to the dye/protein as a way to more directly speak on the organization of life essential building blocks into structures that are observed in nature.
11. Figure 5b: It is difficult to visually observe whether increased localization of newly synthesized lipids are being retained in membranes environments as it appears the newly formed lipid is showing up in many places within the cell (except the nucleus). Can the authors include a co-stain in the membrane environment or some additional analysis to confirm actual membrane co-localization?
12. Figure 5c, 5d, 5e panels. The x and y axis are not clearly marked. When a μM or nmol value is given, would be helpful to the reader to indicate what product or substrate this is referring to. This information provided in the text but is hard to keep track of when one goes to the actual figure.
13. One minor suggestion in the supporting information. The supporting figures/tables/schemes called out in the main text are not easy to navigate to in the supporting materials and are coupled with a lot of other information. It is greatly appreciated that the authors are providing additional information with each of those figures which is helpful, but the authors may want to consider making these called out supp. figures easier to find by grouping them together.

Reviewer #2

(Remarks to the Author)

In their manuscript, Ji et al. explore a novel photocatalytic approach for the non-enzymatic synthesis of natural lipids in both artificial and living cells. The authors demonstrate that visible-light-driven photoredox chemistry enables the radical-mediated coupling of hydrocarbon tails to polar lipid precursors, forming biologically identical lipids. They further show that this photoredox lipid ligation (PLL) strategy can drive de novo vesicle formation, growth, and division, mimicking protocell-like behavior. Additionally, they introduce a unique concept where RNA aptamers can activate photocatalysts, providing a direct link between nucleic acids and abiotic lipid metabolism. Finally, the authors extend their findings to living cells, demonstrating that light-mediated lipid synthesis can generate bioactive lipids, such as ceramides and diacylglycerols, triggering key signaling pathways like apoptosis and PKC activation.

While the study presents an innovative and promising approach with potential applications in synthetic biology, artificial cell systems, and lipid metabolism research, several aspects require further refinement.

1. Please edit the citations of supporting information in the main text according to their order of appearance.
2. The article primarily focuses on the feasibility of the photocatalytic reaction but does not fully assess the cytotoxicity of the

photocatalysts (Eosin Y, Rhodamine B). It is recommended to include: 1) Long-term (>24 h) cell viability assays. 2) The correlation between light intensity and cytotoxicity to evaluate whether cells are affected by photodamage.

3. The article does not provide the quantum yield of the photocatalytic reaction, which is an important parameter in photochemical studies. Could the quantum yield from previous photocatalytic lipid synthesis studies be used as a reference?

4. Figure 5b demonstrates the synthesis of lipids on the cell membrane. However, it lacks adequate negative controls, such as the removal of the photocatalyst or the absence of light exposure, to determine whether lipid modification still occurs and to rule out the possibility of non-specific adsorption. Besides, could the author explain why the red signal lights up the whole cytoplasm, not just the cell membrane?

5. It is recommended to incorporate additional methods, such as radioactive labeling, to further verify the distribution of lipids within the cells. Alternatively, other fluorophore labeling methods, such as using NBD-labelled acyl chain precursors (NATURE COMMUNICATIONS, 2020, 11:4317), could be used to track the distribution of phospholipids.

6. The authors emphasize the advantages of non-enzymatic natural lipid synthesis, but a more detailed comparison with existing lipid synthesis methods (such as chemical esterification and acylation) could be included. This should cover aspects like synthesis efficiency, substrate scope, and biocompatibility to better highlight the breakthrough of this approach. For example, Liu et al., 2020 (Nat. Chem.), which is frequently cited in the text, also reported enzyme-free lipid synthesis. How does this study offer advantages in terms of substrate scope, reaction conditions, and biocompatibility?

7. Please discuss the limitation of this study.

8. Enhance the discussion section by exploring future applications in greater depth. For example, can this method be applied to more complex lipid systems? Can it be integrated with other photocatalytic systems, such as nanocatalysts or MOFs, to improve efficiency? Additionally, further applications in living systems should be considered, can this approach be tested in organoids or mouse models?

Reviewer #3

(Remarks to the Author)

Reviewer #4

(Remarks to the Author)

This manuscript presents an interesting approach to lipid synthesis. It demonstrates the abiotic formation of certain natural lipids in water using photoredox chemistry from acrylate derivatives of lysophospholipids. The authors show how this method enables vesicle formation, growth, and budding and discuss its potential applications in living cells. Particularly interesting is the discovery that RNA aptamers can drive lipid synthesis. The study is well-executed and clearly discusses its broader implications. Overall, this work will undoubtedly interest lipid researchers across multiple disciplines.

A few comments requiring attention:

- How easily can the acrylate derivatives of lysophospholipids, which are the substrates of all reactions described, be generated? It would be helpful if the authors could comment on the synthesis of these substrates.
- Was any attempt made to synthesise glycerophospholipids with headgroups other than choline, e.g., inositol, glycerol, serine, ethanolamine, etc?
- Line 68: A short description of the choice of N-hydroxy phthalimide (NHPI) activation on the fatty acid would be helpful. Indeed, there are other options – but why is this one chosen?
- Figure 1(a) and line 69: There is no specific mention as to whether the classes of lipids synthesised are saturated or unsaturated (and whether the presence of double bonds affects the reactivity) and/or whether these are enantiopure. Surely, this is an important point to make.
- Figure 1(a): The leaving group under the reaction scheme arrow shown as the cartoon headgroup is not immediately apparent. This should be shown as a chemical structure or abbreviation (if at all) in this general scheme.
- Line 75: It is not immediately obvious how Figure 1d relates to the statement made here. Perhaps some labelling in Figure 1d might help.
- Line 92: 'relatively benign' – I understand what the authors are trying to say, but I don't think this should be used in this context.
- Line 107: 'Representative example'? Is it really, though? The chain lengths are not markedly different. Some explanation on choice would be helpful. The number/lettering of compounds also needs to be inserted into the Figure caption – it is currently absent, so the compounds in Fig 2c are not explicitly referred to.
- Line 113: Wavelength of green light? In several previous sentences, green and blue light are given a wavelength. Why not here?
- Line 140/141: Now, saturation and unsaturation are mentioned, but this is not mentioned earlier—why? This also raises the question of the choice of lipid chain—why were certain chain lengths selected over others, with varying degrees of

saturation? There seems to be no clear pattern here.

Line 200: Green LED light—wavelength? This has been specified for other coloured LED lights throughout and green light previously.

- Line 238: 'no POPC synthesis' poor English. Alter to 'POPC was not observed' or something like that.

- References: the formatting of the references seems incorrect throughout (i.e there is no comma after the journal throughout + date is in the wrong position).

- Line 69 to Line 146: Some text restructuring may benefit the reader. For example, Sphingolipids are mentioned in line 69 but are not explicitly explained until later in line 147

Version 1:

Reviewer comments:

Reviewer #1

(Remarks to the Author)

All of my comments have been fully and satisfactorily addressed by the authors.

Reviewer #2

(Remarks to the Author)

I have no further questions. The authors did a good job in revision.

Reviewer #3

(Remarks to the Author)

Reviewer #4

(Remarks to the Author)

I am satisfied with the authors' responses and improvements to the manuscript. I recommend publication of the manuscript in its current state.

Responses to the Reviewer Comments:

Reviewer #1 (Remarks to the Author):

Ji et al. report the development of a photoredox system for assembly of lipids under biocompatible conditions. Their method is called Photoredox Lipid Ligation (PLL) and enables the spatiotemporal control of lipid synthesis. This is achieved via visible light-mediated photoredox activation of N-acyloxy phthalimide esters using eosin Y as a photocatalyst for radical mediated coupling of hydrocarbon tails to polar single chain precursors. The resulting products are newly generated biologically relevant lipids. The authors first demonstrate the broad applicability of this method by synthesizing several different lipid products that vary in lipid tail and head group showcasing the versatility of this method. The authors next show the ability to generate vesicles from these photocatalytically-generated lipids and then follow this up with the use of DNA/RNA-nucleic acid binding dye conjugates that can achieve similar lipid products as their eosin Y method. Finally, the authors apply this labeling approach in living cells by assembling lipid products such as azide-labeled phospholipids, ceramide, and diacylglycerols and monitored the implications of these newly synthesized lipids visually and functionally through cell killing and/or protein activation. In the process, the authors also show that the lipid pre-cursors they use limit unwanted side effects compared to native precursors (e.g. acrylamide-modified sphingosine vs sphingosine) and/or the products they generate can achieve cell permeability in cases where the native products themselves cannot (in the case of ceramide). This highlights the potential utility of PLL for being able to better understand what lipids are doing in living cells without hindrance of background activity or lack of access to native environments.

Overall, this is an exciting work in the field of lipid biology. The ability to generate lipid products in live cells with exquisite spatiotemporal control and monitor their biological functions is an important methodology needed for better understanding how lipids work. Investigation into protein and sugar biomolecule function have long benefited from such tools and our ability to understand their biological functions in isolation or in the larger context of the surrounding cellular environment have greatly benefited as a result. This PLL approach further opens the door for similar studies to be performed on lipid biomolecules. In addition to basic biology pursuits, the ability to generate vesicles with this approach and study impact of lipid content on vesicle shape, structure, dimension could be a huge boost to building artificial vesicles with physiological/therapeutic importance.

Below are some comments to address for helping to improve the manuscript.

[Authors reply] We appreciate the positive comments from the reviewer. We have carefully considered and addressed the concerns raised, as detailed below.

Comment 1. It could be beneficial to show an image or schematic in the main or supporting text that highlights the coupling mechanism. While N-acyloxy phthalimides have been widely used in other synthetic contexts, it would be helpful to introduce how that approach is being leveraged here. In particular, it would be helpful to the reader to highlight how this method compares to current/conventional in vitro lipid synthesis approaches (e.g. a figure showing past or previous assembly methods vs. the assembly method in this new work).

[Authors reply] We thank the reviewer for their suggestion. In response, we have added a new figure (Supplementary Fig. 5, also illustrated below) that provides a detailed schematic of the coupling mechanism utilizing N-acyloxy phthalimides to synthesize phospholipids. Additionally, we have significantly expanded our discussion of previously reported synthetic methodologies for natural phospholipid generation (as also requested by Reviewer #2, comment 6), providing details on their reaction conditions, yields, substrate scope, and biocompatibility, and highlighting the limitations of each approach (Supplementary Fig. 1).

We have modified the main text as below:

(original manuscript)

As an alternative to enzymatic synthesis, several studies have attempted to generate natural lipids, or very close analogs, abiotically in water⁸⁻¹⁴. Previous approaches to abiotically generate natural lipids in water have used acylation chemistry to mimic the enzymatic synthesis of lipids¹²⁻¹⁴. For instance, recent work has shown that lysophospholipids can be acylated in alkaline media to create natural lipids¹⁴. However, these methods have suffered from low yields, limited substrate scope, and the inability to generate membrane lipids under physiologically relevant conditions.

(revised manuscript)

As an alternative to enzymatic synthesis, several studies have attempted to generate natural lipids, or very close analogs, abiotically in water⁸⁻¹⁴. Previous approaches to abiotically generate natural lipids in water have used acylation chemistry to mimic the enzymatic synthesis of lipids¹²⁻¹⁴. For instance, recent work has shown that lysophospholipids can be acylated in alkaline media to create natural lipids¹⁴, but this process requires several hours and provides high yield only under basic conditions. Under more physiologically relevant conditions such as in PBS (pH 7.4) or HEPES buffer (pH 7.5), the reaction yield was negligible (<1%), which makes this approach interesting for prebiotic chemistry, but limits its applicability to biological systems (Supplementary Fig. 1a). Another example of abiotic lipid synthesis involves a chemoselective strategy for in situ ceramide formation¹³, which employs cell-permeable ligation partners to generate ceramides in PBS (pH 7.4, 37 °C). While this method operates under physiological conditions, ceramide formation requires several hours and the method is restricted to ceramide synthesis, limiting its broader utility for the production of diverse lipid species (Supplementary Fig. 1b).

Supplementary Fig. 1. Abiotic synthesis of natural membrane lipids in water via photoredox lipid ligation (PLL). (A) Enzyme-free synthesis of natural phospholipids under prebiotically relevant alkaline conditions. (B) Traceless synthesis of ceramide by acylation of sphingosine in living cells. (C) Unified strategy for the synthesis of a wide range of natural lipids in artificial and living cells through PLL (this work).

(original manuscript)

The mechanism of PLL suggests flexibility in the type of lipids that can be generated, provided that the target natural lipid possesses a fatty acyl tail.

(revised manuscript)

The mechanism of PLL (Supplementary Fig. 5) suggests flexibility in the type of lipids that can be generated, provided that the target natural lipid possesses a fatty acyl tail.

Supplementary Fig. 5. Proposed mechanism for photoredox lipid ligation based on prior literature^{8,9}.

Comment 2. Lines 90-93. What is referred to by recent studies? Is this referring to the paper cited in previous sentences or other papers. Would be good to clarify.

[Authors reply] We thank the reviewer for their careful reading. We have clarified in the text that the “recent studies” refer to references 22 and 23.

(original manuscript)

Recent studies have demonstrated that this conjugation works in aqueous conditions and depending on the choice of photocatalysts, can be conducted with relatively benign visible light (>500 nm excitation), which is an important prerequisite for biocompatibility.

(revised manuscript)

Recent studies^{22,23} have demonstrated that this conjugation works in aqueous conditions and, depending on the choice of photocatalysts, can be conducted with relatively benign visible light (>500 nm excitation)²⁷, which is an important prerequisite for biocompatibility.

Comment 3. Figure 2 panel b. What reaction yields does the condition optimization screen connect to? The yields for 3a? Would be good to more clearly highlight this in the panel and legend.

[Authors reply] We apologize for the confusion and confirm that the reaction yields in Figure 2b refer to compound **3a**. In the revised manuscript we have updated both the panel and the caption of Figure 2b as follows:

(original manuscript Figure 2b)

b

Entry	Deviation from standard conditions ^a	Yield (%) ^b
1	None	95 ^c
2	No light	trace
3	No eosin Y	trace
4	Decyl-NAH instead of BNAH	97
5	BuNAH instead of BNAH	80
6	NADH instead of BNAH	trace
7	Ascorbate instead of BNAH	trace
8	DIPEA instead of BNAH	trace
9	RhodB instead of eosin Y	92
10	DMEM (no phenol red) instead of PBS	88

(revised manuscript Figure 2b)

b

Entry	Deviation from standard conditions ^a	Yield 3a (%) ^b
1	None	95 ^c
2	No light	trace
3	No eosin Y	trace
4	Decyl-NAH instead of BNAH	97
5	BuNAH instead of BNAH	80
6	NADH instead of BNAH	trace
7	Ascorbate instead of BNAH	trace
8	DIPEA instead of BNAH	trace
9	RhodB instead of eosin Y	92
10	DMEM (no phenol red) instead of PBS	88

(original manuscript Figure 2b caption)

Reaction condition optimization using different reductants and photocatalysts as well as controls.

(revised manuscript Figure 2b caption)

Reaction condition optimization using different reductants and photocatalysts (and relative controls) for the synthesis of phospholipid **3a**.

Comment 4. Have the authors tested the reaction in the presence of free fatty acids or other lipid-like molecules to get a sense of the reaction selectivity in the presence of other lipid biomolecule/building blocks? Presumably the reaction would be very selective, but might be good to showcase this with an in vitro experiment as part of the build up story to the utility of this method.

[Authors reply] We appreciate the reviewer's suggestion. To assess the reaction selectivity in the presence of other lipid-like molecules, we tested the reaction in the presence of palmitic acid and cholesterol (results shown in Supplementary Table 4, also reported below). In both cases, we observed similar yields compared to the reaction performed in the absence of any additional lipid, indicating that the reaction proceeds efficiently without significant interference from these lipid biomolecules. We have revised the main text to include these results as below.

(revised manuscript)

We also assessed the efficiency of the PLL reaction in the presence of lipid molecules commonly found in membranes, such as palmitic acid and cholesterol, and observed yields comparable to those obtained in the absence of additional lipids, further supporting the selectivity of PLL (Supplementary Table 4).

Supplementary Table 4. Synthesis of POPC in the presence of palmitic acid or cholesterol

Reaction conditions	standard condition	Standard condition + 1 mM palmitic acid	Standard condition + 1 mM cholesterol
HPLC Yield of POPC (%)	89	92	90

Comment 5. To address the high-yield being driven by product self-assembly, can the authors also perform the reaction in the presence of a suitable detergent to show that disruption or prevention of micelles/emulsion droplets prevents high yields?

[Authors reply] We appreciate the reviewer's suggestion. To assess whether product self-assembly contributes to the high yield obtained for PLL, we performed the reaction in the presence of SDS (40 mM), a detergent known to disrupt micelles and emulsion droplets. Under these conditions, we observed minimal product formation, suggesting that self-assembly plays a crucial role in driving the reaction efficiency. We added a sentence in revised manuscript.

(revised manuscript)

Furthermore, performing the reaction in the presence of 40 mM SDS resulted in minimal product formation (less than 5%), indicating that self-assembly is essential for efficient reaction progress (Supplementary Fig. 4).

Supplementary Fig. 4. HPLC-ELSD traces of POPC synthesis in the presence (A) and absence (B) of 40 mM SDS. Blue traces correspond to the reaction at 0 min (*bottom*), brown traces represent the reaction after 30 min of green light (525 nm) irradiation (*top*).

Comment 6. Figure 2c. Can the authors address why the lower yields for Compounds 3f, 3g, and others in the text.

[Authors reply] We thank the reviewer for their question. Regarding compound 3f, we believe the lower yield obtained is primarily due to the significantly higher cmc of the lipid precursors used for the synthesis of 3a. For comparison the cmc for related 16:0 Lyso PA is 0.54 mM, which is much higher than the cmc for 16:0 Lyso PC, which is 0.005 mM. A higher cmc would likely lead to less efficient mixed micelle formation and thus to lower yields. Regarding compounds 3g and

3h, our hypothesis is that the required acrylamide lipid precursors are less electron deficient, and therefore less reactive, compared to the acrylate ester. This lower reactivity might result in a reduced capacity to trap the alkyl radicals generated, causing a lower reaction yield. Finally, for compound **3i**, the limited water solubility of all precursors, even with the addition of extra detergent, leads to the formation suspended particles, which likely hinders efficient reagent mixing, contributing to lower yield. We apologize for any confusion and hope this explanation clarifies the observed differences in yields. Additionally, to address comment 14 from reviewer #4, we have relocated the two sentences describing sphingolipids (line 146 to 149 of the original manuscript) to an earlier section in the main text.

In response to the reviewer's comment, we have incorporated our hypothesis into the manuscript, as outlined below:

(original manuscript)

We were able to show that alternative lipid head groups could also be used. For instance, PLL yielded diacylglycerol (**3i** in Fig. 2c) and phosphatidic acid (**3f** in Fig. 2c) in aqueous conditions, albeit in lower yields compared to phosphatidylcholines possibly due to the relatively poor solubility of the precursors. Sphingolipids are a class of lipids highly enriched in the nervous system and are involved in several biological processes and diseases³⁰. Sphingolipids have an *N*-acyl fatty tail as opposed to phospholipids which typically possess hydrocarbon tails appended through *O*-acylation. To generate sphingolipids, we synthesized the acrylamide-containing lipid precursors **1e** and **1f**. Using a similar PLL approach, we were able to assemble sphingomyelin (**3g** in Fig. 2c), a major constituent of myelin sheaths, and ceramide (**3h** in Fig. 2c), a signaling sphingolipid involved in apoptosis³¹.

(revised manuscript)

We were able to show that alternative lipid head groups could also be used. For instance, PLL yielded diacylglycerol (**3i** in Fig. 2c) and phosphatidic acid (**3f** in Fig. 2c) in aqueous conditions, albeit with lower yields compared to phosphatidylcholines. We attribute the lower yield of compound **3i** primarily to the limited water solubility of its precursors which, even with additional detergent, led to the formation of insoluble suspended particles with presumably less efficient reagent mixing. We believe the lower yield observed for phosphatidic acid **3f** may be due to the expected higher critical micelle concentration (cmc) of the 16:0 Lyso PA derivative (parent 16:0 Lyso PA cmc is 0.54 mM) compared to that of the 16:0 Lyso PC derivative (parent 16:0 Lyso PC cmc is 0.005 mM)³¹, which may hinder co-localization of the starting materials in self-assembled structures. To generate sphingolipids, we synthesized the acrylamide-containing lipid precursors **1e** and **1f**. Using a similar PLL approach, we were able to synthesize sphingomyelin (**3g** in Fig. 2c), a major constituent of myelin sheaths, and ceramide (**3h** in Fig. 2c), a signaling sphingolipid involved in apoptosis³². The lower yields observed for sphingomyelin **3g** and ceramide **3h** may be explained by the lower electron deficiency of acrylamide-derived precursors compared to acrylate esters, which likely results in decreased reactivity, and limits their ability to capture the generated alkyl radicals.

Comment 7. Line 174. Figures 3a-3c are very quickly glossed over in the text. Would be helpful to the reader to discuss these panels and the kinetics of reaction.

[Authors reply] We thank the reviewer for the insightful comment. In response, we have revised the manuscript to include a more detailed discussion of Figures 3a–3c. Furthermore, we have revised the manuscript to clarify our rationale for using POPC instead of OPPC (please see also our response to comment 9).

(revised manuscript)

Given that **3a** and **3b** (POPC) provided comparable synthetic yields, and that POPC is more prevalent in nature, we selected POPC for further characterization of PLL (Fig. 3a). HPLC-ELSD analysis (Fig. 3b) revealed that POPC formation in PBS under 30 minutes of green light irradiation (525 nm) yielded a clear product peak, whereas no product was detected in the dark. We then conducted kinetic studies under the same conditions, observing a rapid depletion of precursors **1b** and **2b** accompanied by a concomitant increase in POPC, reaching completion within 1 minute (Fig. 3c). These results collectively underscore the efficiency of the photochemical synthesis protocol employed for POPC.

Comment 8. Regarding kinetics, can the authors gage how quickly the vesicles form after synthesis? Is this on par with typical/physiological vesicle formation kinetics? Along these lines, how does vesicle formation and shapes in Figure 3d and 3i compared to formation of vesicles from a native lipid precursor that is simply added together to form vesicles. This would be useful to get a sense of how different or similar vesicle formation is to normal vesicle formation through this synthetic approach. Furthermore, are the interesting vesicle shapes and structures generated through this approach unique to this method? If so, this could possibly open new avenues for spatiotemporal control of vesicle shapes and/or other biological/therapeutic applications.

[Authors reply] Based on observation of de novo vesicle formation on the glass slide, it usually takes several minutes before the vesicles emerge, which is on par with typical vesicle formation during direct hydration. The shapes of de novo formed POPC vesicles on the glass slide (Supplementary Fig. 9C) are similar to the shapes of vesicles formed de novo in a glass vial (Supplementary Fig. 9D) as well the shapes of vesicles obtained by direct hydration of POPC (Supplementary Fig. 9E). Therefore, while we provide a new method for spatiotemporal control of vesicle formation, we currently do not have evidence for whether our method leads to novel vesicle shapes/structures/assembly kinetics. We included a statement in the main text as below.

Supplementary Fig. 9. (C-E) Comparison of POPC vesicle shapes by different formation methods including de novo formation on glass slide (C), de novo formation in glass vial (D), and direct hydration of POPC with PBS buffer (E). Scale bar, 20 μm .

(revised manuscript)

The shape of de novo formed vesicles on the glass slide are similar to vesicles formed by direct hydration of dried POPC (Supplementary Fig. 9).

Comment 9. Figure 3. The back and forth between POPC and OPPC is a bit difficult to follow in the text. Suggest to either stick with one vesicle type in the figure or more clearly call out the vesicle types in the figure images and/or provide more clear rationale or commentary for why moving between POPC and OPPC.

[Authors reply] We thank the reviewer for pointing out the inconsistency. We have revised the manuscript to clarify our rationale for using POPC instead of OPPC. In the updated version, the main text now provides an explanation for our choice (see below), and Figure 3 has been modified for consistency. All panels now present data exclusively for POPC. Specifically, the original panels 3f–3h, which previously showed OPPC data, have been replaced with the corresponding POPC data. Additionally, we have moved the original Figure 3h to the supporting information (Supplementary Fig. 9), as we realized it provides information that is redundant with Figure 3e.

(revised manuscript)

Given that **3a** and **3b** (POPC) provided comparable synthetic yields and that POPC is more prevalent in nature, we selected POPC for further characterization of PLL (Fig. 3a).

Fig. 3. **f** A representative cryoEM image of vesicles formed during the synthesis of POPC **3b** by PLL confirming the presence of a phospholipid bilayer structure. Scale bar, 20 nm. **g, h** Bright field (**g**) and fluorescence (**h**) microscopy images demonstrate that in situ generation of POPC vesicles in the presence of fluorescent protein mCherry leads to spontaneous protein encapsulation. The unencapsulated protein was removed using Ni-NTA beads before imaging vesicles by confocal microscopy. Scale bar, 20 μm .

Comment 10. This reviewer found it difficult to follow the progression from figure 3 to figure 4 and then to figure 5. Seems like there is a more natural progression from figure 3 to figure 5 and that, by inserting figure 4 in between these two sections, there seems to be a deviation from the overall story. The use of DNA/RNA-photocatalyst complexes is an interesting concept to induce lipid and vesicle formation but comes off as just another type of photocatalyst that can be used to generate lipids through this approach rather than an advancement on origin of life study capabilities. As a suggestion to help with the flow of this section, perhaps the authors can determine whether DNA can be encapsulated within the newly generated vesicles similar to the dye/protein as a way to more directly speak on the organization of life essential building blocks into structures that are observed in nature.

[Authors reply] We apologize for the lack of smooth transitions between different sections of the manuscript and appreciate the reviewer's feedback. We have chosen to maintain the current order of the figures as we believe the DNA/RNA section (Figure 4) naturally follows the discussion on membrane growth and division (Figure 3), given the likely relevance of vesicle growth/division in prebiotic chemistry and synthetic cells. Additionally, this structure ensures a clear distinction between *in vitro* and *in cellulo* experiments. We have revised the text to improve the transition between these sections and strengthen the overall narrative. We hope these revisions enhance the readability and logical flow of the manuscript.

Furthermore, to determine whether DNA can be encapsulated within the newly generated vesicles, we performed an additional experiment in which newly formed vesicles were diluted 10-fold and imaged by confocal microscopy. Dilution is expected to affect the external dye brightness but not affect the internalized dye concentration as it is trapped in vesicles. We observed that fluorescence is indeed localized within the vesicles, demonstrating that a portion of the DNA is encapsulated during vesicle formation. This data has now been included in the supplementary information (Supplementary Fig. 17F, also reported below).

Supplementary Fig. 17. (F) Representative microscopy images of a vesicle formed during the synthesis of POPC in the presence of 15 μ M TOTO-1 and 10 ng/mL DNA. Bright field image (*up*) and fluorescence microscopy image (*down*) show that DNA is encapsulated inside the de novo formed vesicle. Scale bar, 10 μ m.

The main text was revised in original manuscript line 238 as follows:

(revised manuscript)

Furthermore, after diluting the newly formed vesicles 10-fold and imaging them by confocal microscopy, we observed fluorescence within the vesicles, demonstrating that a portion of the DNA is encapsulated during vesicle formation (Supplementary Fig. 17). This finding underscores how vesicles generated by PLL can effectively compartmentalize essential biomolecules, a key requirement for life-like structural organization.

Comment 11. Figure 5b: It is difficult to visually observe whether increased localization of newly synthesized lipids are being retained in membranes environments as it appears the newly formed lipid is showing up in many places within the cell (except the nucleus). Can the authors include a co-stain in the membrane environment or some additional analysis to confirm actual membrane co-localization?

[Authors reply] We thank the reviewer for the suggestion. We believe this was due to the internalization of lipids which led to subsequent staining of many intracellular structures. Additionally, our previous microscopy protocol used a low magnification objective, making visualization of subcellular structures and membrane staining more difficult. Considering the fast incorporation of phospholipid into cell membranes (*Nat. Chem.* **2022**, *14*, 1078-1085), we decided to change our previous protocol by shortening the time that cells were incubated before the click chemistry labeling steps. The treatment solution was removed after 5 minutes of green light irradiation, and the HeLa cells were washed 3 times before adding the DBCO-Fluor594 (note: we also modified our protocol in the corresponding supporting information). The shorter incubation minimizes synthetic phospholipid internalization, making it easier to observe lipid localization to subcellular structures. In addition, to better observe the fluorescence localization, we imaged the cell fluorescence signal by 60x magnification (versus 40x in the previous manuscript version). We believe this new data clearly shows that synthesized lipids are retained on the cell membranes, most notably the plasma membrane.

(original manuscript Figure 5b)

·
·
·

(revised manuscript Figure 5b)

Comment 12. Figure 5c, 5d, 5e panels. The x and y axis are not clearly marked. When a μM or nmol value is given, would be helpful to the reader to indicate what product or substrate this is referring to. This information provided in the text but is hard to keep track of when one goes to the actual figure.

[Authors reply] We thank the reviewer for their helpful suggestion. In response, we have revised Figures 5c-5e to include clearly labeled x and y axes. We now indicate whether the reported values correspond to substrates or products directly on the figure panels, as detailed below.

(original manuscript Figure 5c-5e)

(revised manuscript Figure 5c-5e)

Comment 13. One minor suggestion in the supporting information. The supporting figures/tables/schemes called out in the main text are not easy to navigate to in the supporting materials and are coupled with a lot of other information. It is greatly appreciated that the authors are providing additional information with each of those figures which is helpful, but the authors may want to consider making these called out supp. figures easier to find by grouping them together.

[Authors reply] We thank the reviewer for their suggestion. We have reorganized the supporting information by grouping together figures related to the same experiment to improve clarity.

Reviewer #2 (Remarks to the Author):

In their manuscript, Ji et al. explore a novel photocatalytic approach for the non-enzymatic synthesis of natural lipids in both artificial and living cells. The authors demonstrate that visible-light-driven photoredox chemistry enables the radical-mediated coupling of hydrocarbon tails to polar lipid precursors, forming biologically identical lipids. They further show that this photoredox lipid ligation (PLL) strategy can drive de novo vesicle formation, growth, and division, mimicking protocell-like behavior. Additionally, they introduce a unique concept where RNA aptamers can activate photocatalysts, providing a direct link between nucleic acids and abiotic lipid metabolism. Finally, the authors extend their findings to living cells, demonstrating that light-mediated lipid synthesis can generate bioactive lipids, such as ceramides and diacylglycerols, triggering key signaling pathways like apoptosis and PKC activation.

While the study presents an innovative and promising approach with potential applications in synthetic biology, artificial cell systems, and lipid metabolism research, several aspects require further refinement.

[Authors reply] We appreciate the reviewer's supportive feedback and have addressed the points raised, as detailed below.

Comment 1. Please edit the citations of supporting information in the main text according to their order of appearance.

[Authors reply] We thank the reviewer for bringing this to our attention. We have now reviewed the supporting information citations and ensured that they appear in the main text according to their correct order of appearance.

Comment 2. The article primarily focuses on the feasibility of the photocatalytic reaction but does not fully assess the cytotoxicity of the photocatalysts (Eosin Y, Rhodamine B). It is recommended to include: 1) Long-term (>24 h) cell viability assays. 2) The correlation between light intensity and cytotoxicity to evaluate whether cells are affected by photodamage.

[Authors reply] We appreciate the reviewer's concern regarding cytotoxicity. To address this, we conducted a long-term viability study in which cells were treated with varying concentrations of eosin Y (5, 10, 25, 50, 100 μM) and incubated them in darkness for 60 hours. A reduction in cell viability was observed only at the highest concentration (100 μM). However, in the PLL reaction performed in cells, the concentration of eosin Y used was less than 10 μM , which did not affect viability. Since Rhodamine B was not used in cell studies, we did not include any additional cytotoxicity controls for Rhodamine B to the manuscript. Furthermore, we examined the correlation between light intensity and cytotoxicity to assess potential photodamage. Cells were exposed to green light irradiation at 10 W (the power used in our cell study experiments) and 18 W for five minutes (the irradiation time used in our experiments), followed by a 60-hour incubation. The results showed minimal cytotoxic effects even at 18 W, supporting the feasibility of our approach. We have included these results in the revised manuscript.

(revised manuscript)

Furthermore, we conducted long-term viability studies by varying eosin Y concentrations and green light intensity, observing in both cases negligible cytotoxicity under the PLL conditions (Supplementary Fig. 21).

Supplementary Fig. 21. Cell proliferation assay (CCK-8) of HeLa cells upon the treatment of eosin Y with indicated concentration followed by incubation for 60 hours (A) or green LEDs (525 nm) with different intensity for five minutes (B). In the PLL cell-based experiments, the

concentration of eosin Y employed was less than 10 μM , and the green LED power used was 10 W.

Comment 3. The article does not provide the quantum yield of the photocatalytic reaction, which is an important parameter in photochemical studies. Could the quantum yield from previous photocatalytic lipid synthesis studies be used as a reference?

[Authors reply] We thank the reviewer for this suggestion. Our reaction proceeds via a mechanism similar to those previously reported (*J. Am. Chem. Soc.* **2020**, *142*, 20143–20151; *J. Am. Chem. Soc.* **1991**, *113*, 9401-9402). Therefore, based on these papers, we estimate that the quantum yield of our reaction should be over 1. We have added a sentence in the revised manuscript.

(revised manuscript)

Based on previous studies^{15,26}, we estimate the quantum yield of PLL should be over 1, which possibly explains the high efficiency of the reaction.

Comment 4. Figure 5b demonstrates the synthesis of lipids on the cell membrane. However, it lacks adequate negative controls, such as the removal of the photocatalyst or the absence of light exposure, to determine whether lipid modification still occurs and to rule out the possibility of non-specific adsorption. Besides, could the author explain why the red signal lights up the whole cytoplasm, not just the cell membrane?

[Authors reply] We thank the reviewer for this comment. Several control experiments were included in the original submission; however, they were detailed in the supplementary information and not explicitly mentioned in the main text. We apologize for the lack of clarity. In the revised manuscript, we have performed additional control experiments and ensured that all controls are appropriately referenced in the main text to improve clarity and accessibility. Our original work included control experiments using untreated cells (Supplementary Fig. 20A), cells treated in absence of light (Fig 2b (*left*), and Supplementary Fig. 20B), as well as cells treated with all the necessary components for PLL except for lysolipid **1b**, and exposed to green light irradiation (Supplementary Fig. 20C). We have now included an additional control performed in absence of photocatalyst (Supplementary Fig. 20D). In all the control experiments performed, no significant fluorescence was observed. In response to this comment and reviewer#1 (comment 11), we changed our cell staining protocol and obtained new images, which clearly shows the red signal primarily localized on the cell membrane. In our previous protocol, the long incubation of the reaction solution likely led to internalization of the synthetic lipids, which contributed to the red signal in the interior of the cell.

Furthermore, we have added a statement in the main text to explicitly mention the control experiments that were conducted.

(original manuscript)

In the absence of light or lysolipid **1b**, significantly less staining was observed (Fig. 5b and Supplementary Fig. 23), likely due to the ability of the single-chain precursors to be washed out more readily than the phospholipid product.

(revised manuscript)

To ensure the robustness of our findings, we performed several control experiments, including imaging untreated cells (Supplementary Fig. 20a), treated cells in the absence of light (Fig. 5b (*left*) and Supplementary Fig. 20b), and treated cells with all the necessary components for PLL under green light (525 nm), except for lysolipid **1b** (Supplementary Fig. 20c), or photocatalyst (Supplementary Fig. 20d). In all controls, significantly less staining was observed (Fig. 5b (*left*) and Supplementary Fig. 20), likely due to the ability of the single-chain precursors to be washed out more readily than the two-chain phospholipid product.

Comment 5. It is recommended to incorporate additional methods, such as radioactive labeling, to further verify the distribution of lipids within the cells. Alternatively, other fluorophore labeling methods, such as using NBD-labelled acyl chain precursors (NATURE COMMUNICATIONS, 2020, 11:4317), could be used to track the distribution of phospholipids.

[Authors reply] We thank the reviewer for this suggestion. In response to reviewer#1 (comment 11), we changed our previous protocol by shortening the incubation duration of the treatment solution. Specifically, after 5 minutes of irradiation, the HeLa cells were washed and then treated by DBCO-Fluor594. We believe our new results clearly show the distribution of the newly synthesized lipid on the cell membranes. Therefore, we respectfully feel it is not necessary to include radioactive labeling experiments. We also note that NBD imaging would not be possible due to the spectral overlap with eosin Y.

Comment 6. The authors emphasize the advantages of non-enzymatic natural lipid synthesis, but a more detailed comparison with existing lipid synthesis methods (such as chemical esterification and acylation) could be included. This should cover aspects like synthesis efficiency, substrate scope, and biocompatibility to better highlight the breakthrough of this approach. For example, Liu et al., 2020 (Nat. Chem.), which is frequently cited in the text, also reported enzyme-free lipid synthesis. How does this study offer advantages in terms of substrate scope, reaction conditions, and biocompatibility?

[Authors reply] We thank the reviewer for this suggestion. In the revised manuscript, we have significantly expanded our discussion of previously reported synthetic methodologies for natural phospholipid generation, providing additional details on their reaction conditions, yields, substrate scope, and biocompatibility, while also highlighting the limitations of each approach.

We have modified the main text as below:

(original manuscript)

As an alternative to enzymatic synthesis, several studies have attempted to generate natural lipids, or very close analogs, abiotically in water⁸⁻¹⁴. Previous approaches to abiotically generate natural lipids in water have used acylation chemistry to mimic the enzymatic synthesis of lipids¹²⁻¹⁴. For instance, recent work has shown that lysophospholipids can be acylated in alkaline media to create natural lipids¹⁴. However, these methods have suffered from low yields, limited substrate scope, and the inability to generate membrane lipids under physiologically relevant conditions.

(revised manuscript)

As an alternative to enzymatic synthesis, several studies have attempted to generate natural lipids, or very close analogs, abiotically in water⁸⁻¹⁴. Previous approaches to abiotically generate natural lipids in water have used acylation chemistry to mimic the enzymatic synthesis of lipids¹²⁻¹⁴. For instance, recent work has shown that lysophospholipids can be acylated in alkaline media to create natural lipids¹⁴, although this process requires several hours and provides high yield only under basic conditions. Under physiologically relevant conditions such as PBS (pH 7.4) or HEPES buffer (pH 7.5), the reaction yield was negligible (<1%), which makes this approach interesting for prebiotic chemistry, but limits its applicability to biological systems (Supplementary Fig. 1a). Another example of abiotic lipid synthesis involves a chemoselective strategy for in situ ceramide formation¹³, which employs cell-permeable ligation partners to generate ceramides in PBS (pH 7.4, 37 °C). While this method operates under physiological conditions, ceramide formation requires several hours and the method is restricted to ceramide synthesis, limiting its broader utility for the production of diverse lipid species (Supplementary Fig. 1b).

Comment 7. Please discuss the limitation of this study.

[Authors reply] We thank the reviewer for this comment. In response, we have revised the manuscript to include limitations and future directions, as detailed below.

(revised manuscript)

While this study demonstrates the efficiency of PLL as a strategy for de novo membrane formation, one key aspect that remains to be explored is whether the encapsulation of the catalyst within vesicles can effectively trigger the reaction from inside the compartments. This would be a crucial step towards developing autonomous synthetic cells capable of internally regulating membrane synthesis and remodeling. Further studies are needed to assess the feasibility of this approach and to investigate the functional properties that may emerge from such an ability. Additionally, future efforts should focus on identifying alternative photocatalysts or reaction pathways that can be activated at higher wavelengths, such as in the near-infrared range, to further enhance the biocompatibility of PLL and possibly open up in vivo applications. Due to synthetic constraints, we only focused on a subset of common phospholipids (e.g. phosphocholine, phosphatidic acid). Expanding this approach to include phospholipids with inositol, serine, or ethanolamine headgroups is an important direction for subsequent work, and we aim to investigate strategies to overcome the associated synthetic challenges.

Comment 8. Enhance the discussion section by exploring future applications in greater depth. For example, can this method be applied to more complex lipid systems? Can it be integrated with other photocatalytic systems, such as nanocatalysts or MOFs, to improve efficiency? Additionally, further applications in living systems should be considered, can this approach be tested in organoids or mouse models?

[Authors reply] We appreciate the reviewer's suggestion. In response, we have broadened the discussion to consider potential future applications in more detail.

(revised manuscript)

Other than nucleic acid binding dyes, we anticipate the employment of photoactivable metal-organic frameworks or nanocatalysts with tunable structures and large surface areas can potentially improve the reaction efficiency for some challenging substrates (**3f**) or more complex lipid synthesis^{49,50}. However, considering the unknown metabolic stability of PLL precursors, it may be challenging to apply PLL for animal model studies, though aforementioned development of near-infrared photocatalysts and more stable PLL precursors would facilitate studies in more complex biological systems.

Reviewer #3 (Remarks to the Author):

[Authors reply] We appreciate the time and effort the reviewer has dedicated to evaluating our manuscript.

Reviewer #4 (Remarks to the Author):

This manuscript presents an interesting approach to lipid synthesis. It demonstrates the abiotic formation of certain natural lipids in water using photoredox chemistry from acrylate derivatives of lysophospholipids. The authors show how this method enables vesicle formation, growth, and budding and discuss its potential applications in living cells. Particularly interesting is the discovery that RNA aptamers can drive lipid synthesis. The study is well-executed and clearly discusses its broader implications. Overall, this work will undoubtedly interest lipid researchers across multiple disciplines.

[Authors reply] We appreciate the positive comments from the reviewer.

Comment 1. How easily can the acrylate derivatives of lysophospholipids, which are the substrates of all reactions described, be generated? It would be helpful if the authors could comment on the synthesis of these substrates.

[Authors reply] We thank the reviewer for this comment. In response, we have added a discussion in the manuscript commenting on the accessibility of these substrates. We believe that all precursors can be synthesized following the procedures detailed in the supplementary information. However, we acknowledge that the acylation of lysophospholipids can sometimes present challenges, particularly requiring extended reaction times. We have added a brief discussion in the revised manuscript to address these considerations and provide further context on the synthesis of these key substrates.

(revised manuscript)

The acrylate derivatives of lysophospholipids can be easily synthesized following established procedures (see Supplementary Information). While acylation typically proceeds efficiently, some lysophospholipids (**1b** and **1c**) may require prolonged reaction time.

Comment 2. Was any attempt made to synthesise glycerophospholipids with headgroups other than choline, e.g., inositol, glycerol, serine, ethanolamine, etc?

[Authors reply] We appreciate the reviewer's interest in the synthesis of glycerophospholipids with alternative headgroups. In this study, we focused on choline-based lipids due to the complexity of the protection and deprotection steps required for other headgroups, as well as the already broad structural diversity we could explore within the phosphocholine series. However, we also tested PLL to obtain lipids with different head groups such as phosphatidic acid (**3f**), ceramide (**3h**), and glycerol-based lipids (**3i**).

Comment 3. Line 68: A short description of the choice of N-hydroxy phthalimide (NHPI) activation on the fatty acid would be helpful. Indeed, there are other options – but why is this one chosen?

[Authors reply] We thank the reviewer for the comment. We have added a short description on the choice of NHPI to our revised manuscript.

(original manuscript)

Photoredox lipid ligation (PLL) between N-hydroxyphthalimide (NHPI) fatty esters and olefin-modified lysolipids forms carbon-carbon bonds, generating natural lipids.

(revised manuscript)

Photoredox lipid ligation (PLL) between N-hydroxyphthalimide (NHPI) fatty esters, one of the most extensively studied precursors for radical coupling^{22,23}, and olefin-modified lysolipids forms carbon-carbon bonds, generating natural lipids.

Comment 4. Figure 1(a) and line 69: There is no specific mention as to whether the classes of lipids synthesised are saturated or unsaturated (and whether the presence of double bonds affects the reactivity) and/or whether these are enantiopure. Surely, this is an important point to make.

[Authors reply] We thank the reviewer for highlighting this point. According to our observations, the presence of unsaturation in the lipid chain does not affect the reactivity of PLL. As a result, both saturated and unsaturated lipids can be efficiently synthesized in high yield under our reaction conditions, and this is further supported by the data presented in Figure 2c (e.g. for compounds **3a** and **3b**), which shows that the double bond is retained and does not affect the reactivity. Furthermore, the acrylate-derived lysophospholipid starting materials used in our procedure are enantiopure, and therefore we infer that the final product remains enantiopure as well.

We have clarified this in the revised manuscript, and included more details on the lipid classes (please see also our response to comment 14).

(original manuscript)

Several lipid classes, including phospholipids, sphingolipids, and diacylglycerols, can be formed in water.

(revised manuscript)

Several lipid classes can be formed in water, including phospholipids (the main structural components of cell membranes, with hydrocarbon tails appended through *O*-acyl alkyl tails), sphingolipids (highly enriched in the nervous system and involved in several biological processes and diseases²⁴, with an *N*-acyl alkyl tail), and diacylglycerols (key intermediates in lipid metabolism, bearing two fatty acyl chains on a glycerol backbone). The presence of unsaturation in the precursors does not interfere with the occurrence of PLL.

Comment 5. Figure 1(a): The leaving group under the reaction scheme arrow shown as the cartoon headgroup is not immediately apparent. This should be shown as a chemical structure or abbreviation (if at all) in this general scheme.

[Authors reply] We thank the reviewer for their suggestion. We have revised Figure 1a to explicitly depict the leaving group as a chemical structure, ensuring greater clarity in the reaction scheme.

(original manuscript Figure 1a)

(revised manuscript Figure 1a)

Comment 6. Line 75: It is not immediately obvious how Figure 1d relates to the statement made here. Perhaps some labelling in Figure 1d might help.

[Authors reply] We thank the reviewer for their suggestion. In response, we have revised Figure 1d to improve clarity by adding additional labels.

Comment 7. Line 92: ‘relatively benign’ – I understand what the authors are trying to say, but I don’t think this should be used in this context.

[Authors reply] We thank the reviewer for their suggestion. In the revised manuscript, we have removed the phrase ‘relatively benign’ to ensure clarity and maintain scientific accuracy.

(original manuscript)

Recent studies have demonstrated that this conjugation works in aqueous conditions and depending on the choice of photocatalysts, can be conducted with relatively benign visible light (>500 nm excitation), which is an important prerequisite for biocompatibility.

(revised manuscript)

Recent studies^{22,23} have demonstrated that this conjugation works in aqueous conditions and depending on the choice of photocatalyst, can be conducted with visible light (>500 nm excitation)²⁷, which is an important prerequisite for biocompatibility.

Comment 8. Line 107: ‘Representative example’? Is it really, though? The chain lengths are not markedly different. Some explanation on choice would be helpful. The number/lettering of compounds also needs to be inserted into the Figure caption – it is currently absent, so the compounds in Fig 2c are not explicitly referred to.

[Authors reply] We thank the reviewer for their thoughtful comment. In our study, we focused on varying lipid classes by modifying headgroups and the degree of unsaturation, as these are key distinguishing factors for lipids. However, we agree with the reviewer's point that chain length is also an important parameter. To enhance clarity, we have removed the word "representative" from the caption and have revised it to include the numbering/lettering of compounds.

(original manuscript Figure 2c caption)

c, Representative examples of natural lipids (phosphatidylcholines, sphingolipids, diacylglycerol) formed by PLL. Additional chemical structures can be found in Supplementary Table 2.

(revised manuscript Figure 2c caption)

c, Examples of natural lipids including phosphatidylcholines (**3b**, **3c**, **3d**, **3e**), phosphatidic acid (**3f**), sphingomyelin (**3g**), ceramide (**3h**), and diacylglycerol (**3i**) formed by PLL. Additional chemical structures can be found in Supplementary Table 3.

Comment 9. Line 113: Wavelength of green light? In several previous sentences, green and blue light are given a wavelength. Why not here?

[Authors reply] We thank the reviewer for their careful reading of the manuscript. We have modified the figure caption to provide the wavelength of light.

(original manuscript)

"under green light for 30 min at room temperature."

(revised manuscript)

"under green light irradiation (525 nm) for 30 min at room temperature."

Comment 10. Line 140/141: Now, saturation and unsaturation are mentioned, but this is not mentioned earlier—why? This also raises the question of the choice of lipid chain—why were certain chain lengths selected over others, with varying degrees of saturation? There seems to be no clear pattern here.

[Authors reply] We apologize if our earlier description created confusion regarding the presence of unsaturation in the lipid alkyl tails. In light of the reviewer's earlier suggestion (comment 4), we now explicitly state in an earlier section of the main text that the presence of unsaturated alkyl chains does not significantly affect PLL (please see our response to comment 4). In this specific context (lines 140/141 of the original manuscript), we discuss Figure 2c, where we highlight that both saturated and unsaturated (including polyunsaturated) lipids can be synthesized through our PLL approach. Regarding chain length, we selected C16/C18 alkyl tails due to their prevalence in natural phospholipids, reflecting the most common chain lengths found in biological membranes.

Comment 11. Line 200: Green LED light—wavelength? This has been specified for other coloured LED lights throughout and green light previously.

[Authors reply] We thank the reviewer for pointing this issue out. We have modified the main text to provide the wavelength of light.

(original manuscript)

“After irradiation with green LED light”

(revised manuscript)

“After irradiation with green light (525 nm)”

Comment 12. Line 238: ‘no POPC synthesis’ poor English. Alter to ‘POPC was not observed’ or something like that.

[Authors reply] We thank the reviewer for their suggestion. We modified the sentence in the revised manuscript as suggested.

(original manuscript)

“In the absence of DNA, no POPC synthesis was observed”

(revised manuscript)

“In the absence of DNA, POPC formation was not observed”

Comment 13. References: the formatting of the references seems incorrect throughout (i.e there is no comma after the journal throughout + date is in the wrong position).

[Authors reply] We thank the reviewer for their comment on our reference formatting. We have carefully revisited *Nat. Commun.* submission guidelines (www.nature.com/ncomms/submit/how-to-submit), and have updated our references accordingly. For instance, we noted that we had inadvertently capitalized all words in the titles, while only the first word should be capital. Furthermore, we confirmed that no comma is required after the journal name and that the year should appear in parentheses following the volume and page range (e.g., *Nature* **344**, 524–526 (1990)). We have now corrected the references thoroughly, and we appreciate the reviewer’s help in improving the consistency of our references.

Comment 14. Line 69 to Line 146: Some text restructuring may benefit the reader. For example, Sphingolipids are mentioned in line 69 but are not explicitly explained until later in line 147.

[Authors reply] We thank the reviewer for this helpful suggestion. To improve clarity for readers unfamiliar with sphingolipids, we have moved the explanation from line 146 and combined to the sentence reported in line 69 of the original manuscript, ensuring that the discussion flows more logically and is easier to follow.

(original manuscript)

Line 69. Several lipid classes, including phospholipids, sphingolipids, and diacylglycerols, can be formed in water.

Line 146. Sphingolipids are a class of lipids highly enriched in the nervous system and are involved in several biological processes and diseases³⁰. Sphingolipids have an *N*-acyl fatty tail as opposed to phospholipids which typically possess hydrocarbon tails appended through *O*-acylation.

(revised manuscript)

Several lipid classes can be formed in water, including phospholipids (the main structural components of cell membranes, with hydrocarbon tails appended through *O*-acyl alkyl tails), sphingolipids (highly enriched in the nervous system and involved in several biological processes and diseases²⁴, with an *N*-acyl alkyl tail), and diacylglycerols (key intermediates in lipid metabolism, bearing two fatty acyl chains on a glycerol backbone).